DOI: 10.1038/s41467-018-07392-7　OPEN

# Structural snapshot of a bacterial phytochrome in its functional intermediate state

Andrea Schmidt[1], Luisa Sauthof[1], Michal Szczepek[1], Maria Fernandez Lopez[2], Francisco Velazquez Escobar[2], Bilal M. Qureshi [1,5], Norbert Michael[2], David Buhrke[2], Tammo Stevens[1], Dennis Kwiatkowski[1], David von Stetten[3,6], Maria Andrea Mroginski [2], Norbert Krauß[4], Tilman Lamparter [4], Peter Hildebrandt[2] & Patrick Scheerer[1]

Phytochromes are modular photoreceptors of plants, bacteria and fungi that use light as a source of information to regulate fundamental physiological processes. Interconversion between the active and inactive states is accomplished by a photoinduced reaction sequence which couples the sensor with the output module. However, the underlying molecular mechanism is yet not fully understood due to the lack of structural data of functionally relevant intermediate states. Here we report the crystal structure of a Meta-F intermediate state of an Agp2 variant from *Agrobacterium fabrum*. This intermediate, the identity of which was verified by resonance Raman spectroscopy, was formed by irradiation of the parent Pfr state and displays significant reorientations of almost all amino acids surrounding the chromophore. Structural comparisons allow identifying structural motifs that might serve as conformational switch for initiating the functional secondary structure change that is linked to the (de-)activation of these photoreceptors.

[1] Charité – Universitätsmedizin Berlin, Corporate Member of Freie Universität Berlin, Humboldt-Universität zu Berlin, and Berlin Institute of Health, Institute for Medical Physics and Biophysics, Group Protein X-ray Crystallography and Signal Transduction, Charitéplatz 1, Berlin, D-10117, Germany. [2] Technische Universität Berlin, Institut für Chemie, Sekr. PC 14, Straße des 17. Juni 135, Berlin, D-10623, Germany. [3] Structural Biology Group, European Synchrotron Radiation Facility, CS 40220 F-38043 Grenoble Cedex 9, France. [4] Karlsruhe Institute of Technology (KIT), Botanical Institute, Fritz-Haber-Weg 4, Karlsruhe, D-76131, Germany. [5] Present address: Division of Biological & Environmental Sciences & Engineering, King Abdullah University of Science and Technology (KAUST), Thuwal, 23955–6900, Saudi Arabia. [6] Present address: European Molecular Biology Laboratory (EMBL), Hamburg Outstation c/o DESY, Notkestrasse 85, Hamburg, D-22607, Germany. These authors contributed equally: Andrea Schmidt, Luisa Sauthof, Michal Szczepek, Maria Fernandez Lopez. Correspondence and requests for materials should be addressed to P.H. (email: peter.hildebrandt@tu-berlin.de) or to P.S. (email: patrick.scheerer@charite.de)

Phytochromes are biliproteins of plants, bacteria and fungi that use red light as a source of information to regulate fundamental physiological processes, such as photosynthesis, seed germination, flowering and shade avoidance[1–3]. The molecular architecture of phytochromes shares a common photosensory core module (PCM), which is composed of PAS (Per/Arnt/Sim), GAF (cGMP phosphodiesterase/adenyl cyclase/FhlA) and PHY (phytochrome-specific) domain[4–6] (Fig. 1a). Despite the differences between the bilin-type chromophores (biliverdin (BV) vs. phytochromobilin) and their covalent binding sites (PAS vs. GAF domain[7]) in bacterial and plant PCMs, the photoinduced processes that interconvert the two parent states Pr and Pfr follow similar pathways with spectrally distinguishable Lumi and Meta intermediates[8,9].

Bacterial phytochromes can be grouped into prototypical and bathy phytochromes, in which the red-absorbing Pr and far-red-absorbing Pfr are the thermodynamically stable states,

respectively[10–13]. The photoconversions between the parent states Pr and Pfr[14] are coupled to downstream signalling processes via the modulation of an attached output module, typically a histidine kinase[15,16]. Coupling starts in the last step of the respective photoconversion with the formation of a Meta intermediate[17] and comprises a secondary structure transition in a sensor-output interconnecting region, the PHY tongue[18,19] (Fig. 1b), which probably represents a reaction pattern common to all phytochromes[18–20]. Thus structural knowledge of the Meta intermediate is a prerequisite for a detailed understanding of the molecular mechanism of the signal transfer from the sensor to the output module (Fig. 1). In a previous study, irradiation of Pfr crystals of the bathy phytochrome PaBphP from Pseudomonas aeruginosa at different temperatures revealed insight into very early photoinduced structural changes of the chromophore in the Lumi-F state, i.e. the E→Z isomerization at the methine bridge C15–C16 between the pyrrole rings C and D of the BV

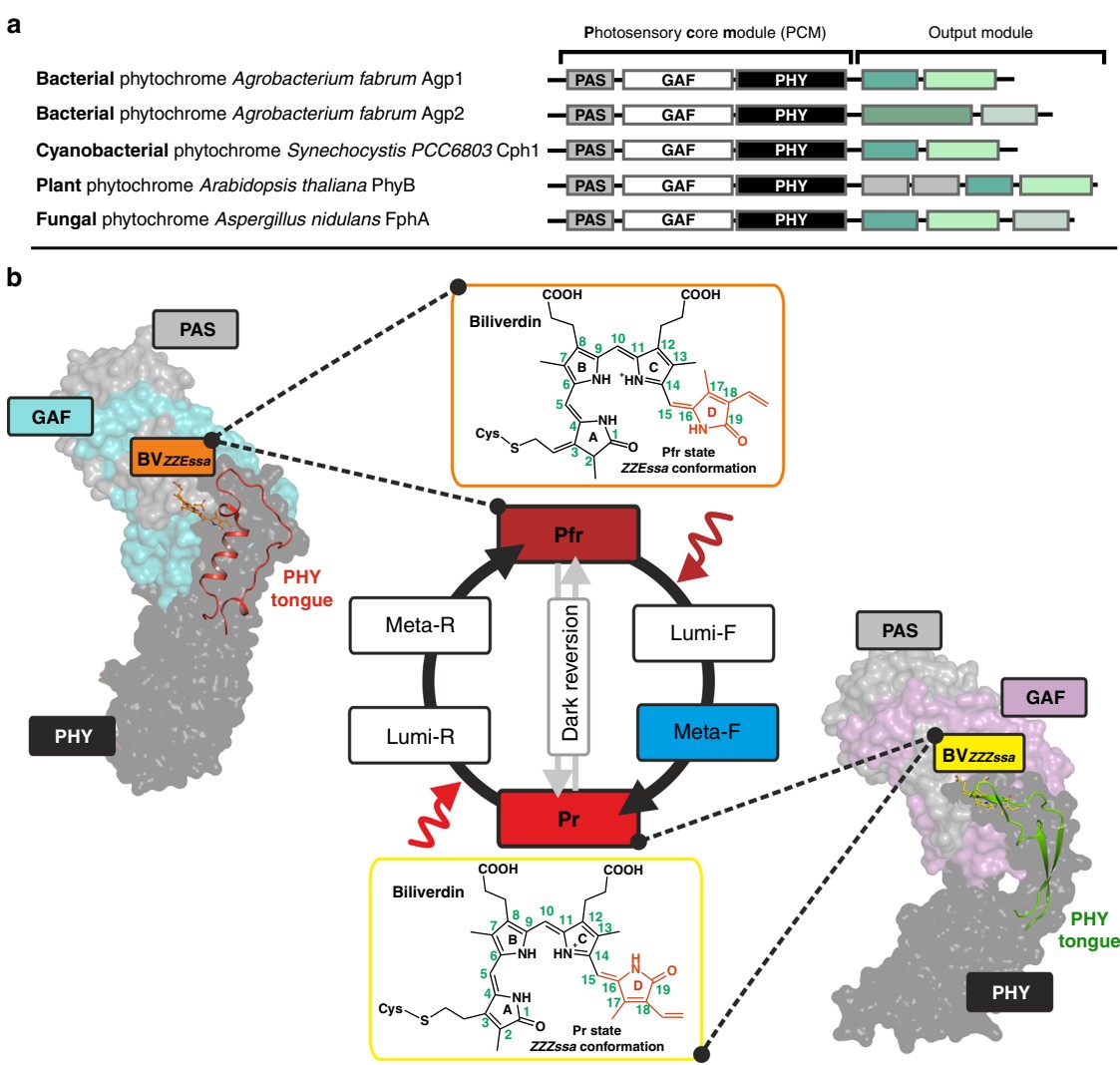

**Fig. 1** Domain arrangement and overview of the phytochrome photocycle. **a** Domain arrangement of different phytochromes. The highly conserved photosensory core module (PCM) is subdivided into PAS, GAF and PHY domains; the C-terminal output modules differ between phytochromes but contain in most cases a histidine kinase (HK) or HK-like module. **b** Overview of the photocycle of phytochromes. For Pfr and Pr states, the overall folds of representative PCM constructs are depicted in surface representation (Pfr state based on wild-type Agp2-PCM: PDB entry 6G1Y, PAS in grey, GAF in cyan, PHY in black; Pr state based on Agp1-PCM$_{SER13}$: PDB entry 5HSQ[30], PAS in grey, GAF in magenta, PHY in black). The secondary structure changes within the PHY tongue (elongated part of the PHY domain) that occur during the last steps of the Pr/Pfr photo-transformation are highlighted in cartoon representation (Pfr: α-helix, loop and coiled structure - red, Pr: β-sheet and β-hairpin - green). The structure of the biliverdin (BV) chromophore in the Pfr and Pr state is shown above and underneath the photocycle scheme, respectively. The chemical structure of BV is represented by only one of their resonance structures each

chromophore[21]. No structural models, however, are yet available for the later intermediates that, instead, have been characterized by vibrational spectroscopy indicating chromophore relaxation and proton transfer steps, in addition to the secondary structure transition of the PHY tongue[17,22–24].

Here we describe the crystal structure of a Meta-F intermediate of the PCM of a bathy phytochrome at 2.16 Å resolution formed during the Pfr→Pr phototransformation (Fig. 2). The intermediate has been generated by irradiating crystals of the parent Pfr state at ambient temperature and identified as Meta-F by resonance Raman (RR) spectroscopy. The conjugate Meta-F and Pfr crystal structures has been obtained from the engineered photoactivatable near-IR fluorescent protein (Agp2-PAiRFP2)[25,26], derived from the bathy phytochrome Agp2 from *Agrobacterium fabrum*. The parent Pfr state of Agp2-PAiRFP2 (2.03 Å resolution) displays the same structure as the wild-type protein determined here as well (2.50 Å resolution) (Fig. 2).

### Results

**Crystal structure of the wild-type bathy phytochrome Agp2-PCM.** Starting point of the present studies was the crystal structure of the PCM fragment of wild-type Agp2 (Agp2-PCM) in the dark adapted Pfr state, which crystallized in the orthorhombic space group $P2_12_12_1$. The structure, solved at 2.5 Å resolution (Fig. 2a; Table 1), shows overall similar topologies to *Pa*BphP-PCM[27], including the characteristic, well-ordered helical conformation of a part of the PHY tongue (Fig. 2a). In contrast to *Pa*BphP-PCM, the tongue region of Agp2-PCM is not directly stabilized by crystallographic contacts to symmetry-related molecules within the crystal. The BV chromophore, β-facially attached to Cys13 via ring A, adopts a ZZEssa stereochemistry (Fig. 2a; Supplementary Figs. 1a, d, g, 2a, d, g). A hydrogen-bonding network around the propionate side chains of rings B and C involves the side chains of Tyr165, Arg211 and His278 and is extended via several ordered water molecules to include also Tyr205, Arg242, Ser245, His248, Ser260 and Ser262 (Fig. 2a; Supplementary Fig. 3). The N-H groups of the rings A, B, C interact with the backbone oxygen of Asp196 and a conserved water molecule (Wat722), whereas the N-H group of ring D forms a salt bridge with the carboxylate group of Asp196. These interactions are conserved in the Pfr state of other bathy phytochromes[27].

The Pfr Agp2-PCM crystals lose their diffraction capability immediately after irradiation with far-red light at ambient temperatures indicating major protein conformational changes in the crystal. We have, therefore, extended the studies to Agp2-PCM variants with substitutions (Supplementary Fig. 4), which do not affect essential structural and mechanistic properties.

**Crystal structure of the Agp2-PAiRFP2 in the Pfr state.** Preliminary crystallization and spectroscopic experiments identified the PCM fragment of the photoactivatable phytochrome Agp2-PAiRFP2 as a promising candidate. Ultraviolet–visible absorption spectra of Agp2-PAiRFP2 compared to the wild-type Agp2-PCM construct, the full-length Agp2 (Agp2-FL) and the prototypical full-length phytochrome Agp1 (Agp1-FL) display the characteristic spectrum of Pfr but an enhanced stability of the final photoproduct state, resulting in slower dark reversion to Pfr compared to Agp2-PCM (Supplementary Fig. 5). The half-lives of the photoinduced states of Agp2-FL, Agp2-PCM and the PCM fragment of Agp2-PAiRFP2 are 170 s[11], 20 s[11] and 233 min[25], respectively. This Agp2-PCM variant includes 24 substitutions, which except for Val244Phe and Ala276Val do not participate in the interaction between the chromophore and chromophore-binding site (Supplementary Figs. 3, 4, 6).

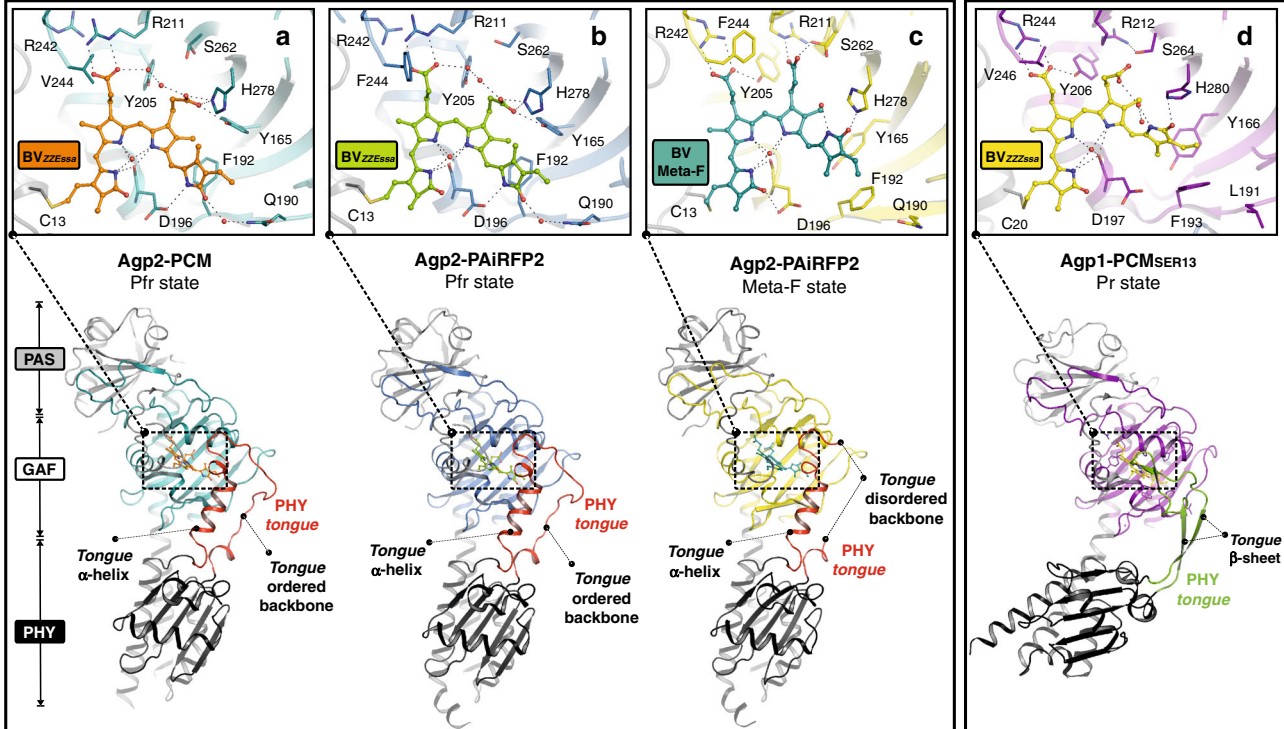

**Fig. 2** Overview of the crystal structures of bacterial phytochromes from *A. fabrum*. Structures (ribbon representation) of the monomers are shown for the dark adapted Pfr state of **a** wild-type Agp2-PCM (PDB entry 6G1Y) and **b** Agp2-PAiRFP2 (PDB entry 6G1Z), **c** a Meta-F sub-state (Mol A) of the Agp2-PAiRFP2 (PDB entry 6G20) and **d** the Pr state of the Agp1-PCM (Agp1-PCM$_{SER13}$ of *A. fabrum*, PDB entry 5HSQ[30]). Close-up view of the chromophore-binding pockets (**a–d**) including BV and amino acid residues within the hydrogen bond network of BV (stick representation) as well as selected water molecules (red spheres). PAS domains are displayed in grey, PHY domains in black and GAF domains are depicted in different colours

**Table 1 Data collection and refinement statistics**

| | Agp2-PCM[a] Pfr state (PDB entry 6G1Y) | Agp2-PAiRFP2[a] Pfr state (PDB entry 6G1Z) | Agp2-PAiRFP2[a] Meta-F state (PDB entry 6G2O) |
|---|---|---|---|
| **Data collection** | | | |
| Beamline | ESRF, ID30A-3[61] | BESSY II (BL14.1)[40] | ESRF, ID23-1[62] |
| Wavelength | $\lambda = 0.9677$ Å | $\lambda = 0.9184$ Å | $\lambda = 0.9724$ Å |
| Space group | $P2_12_12_1$ | $P6_322$ | $P6_322$ |
| Cell dimensions | | | |
| $a, b, c$ (Å) | 74.52, 93.36, 174.42 | 182.38, 182.38, 179.88 | 183.11, 183.11, 179.65 |
| $\alpha, \beta, \gamma$ (°) | 90.0, 90.0, 90.0 | 90.0, 90.0, 120.0 | 90.0, 90.0, 120.0 |
| Resolution (Å) | 45.09–2.50 (2.59–2.50)[b] | 47.73–2.03 (2.14–2.03)[b] | 47.79–2.16 (2.28–2.16)[b] |
| $R_{merge}$ | 0.187 (0.931) | 0.143 (1.495) | 0.138 (1.824) |
| $R_{pim}$ | 0.132 (0.661) | 0.040 (0.417) | 0.031 (0.409) |
| $I/\sigma(I)$ | 8.6 (1.9) | 15.4 (1.9) | 18.3 (1.9) |
| CC1/2 | 0.989 (0.622) | 0.999 (0.592) | 0.999 (0.672) |
| Completeness (%) | 99.9 (99.9) | 100.0 (100.0) | 100.0 (100.0) |
| Redundancy | 5.4 (5.6) | 13.3 (13.6) | 20.1 (20.8) |
| **Refinement** | | | |
| Resolution (Å) | 45.09–2.50 | 47.73–2.03 | 47.79–2.16 |
| No. of reflections | 40755 | 107549 | 90118 |
| $R_{work}/R_{free}$ (%) | 20.1/24.1 | 18.0/21.3 | 20.9/23.2 |
| No. of atoms | | | |
| Protein | 7745 | 7750 | 7682 |
| Ligand/ion | 86 | 244 | 262 |
| Water | 202 | 853 | 311 |
| B-factors | | | |
| Protein | 37.00 | 32.70 | 51.30 |
| Ligand/ion | 25.13 | 45.35 | 68.42 |
| Water | 32.50 | 41.20 | 49.90 |
| R.m.s. deviations | | | |
| Bond lengths (Å) | 0.009 | 0.013 | 0.011 |
| Bond angles (°) | 1.33 | 1.54 | 1.47 |
| Ramachandran plot[c] | | | |
| Favoured (%) | 98.6 | 99.0 | 99.0 |
| Allowed (%) | 1.4 | 1.0 | 1.0 |
| Outlier (%) | 0.0 | 0.0 | 0.0 |

R.m.s. root mean square
[a]One crystal was used
[b]Highest resolution shell is shown in parenthesis
[c]Ramachandran plot calculated by MolProbity[52]

Agp2-PAiRFP2 was crystallized in the dark adapted Pfr state (in the hexagonal space group P6₃22) and the structure was solved at a resolution of 2.03 Å (Table 1). Remarkably, crystal packing and crystallization conditions differ significantly from those of Agp2-PCM. While Agp2-PCM shows a parallel orientation of the two crystallographic monomers, Agp2-PAiRFP2 monomers crystallize in a near anti-parallel arrangement (Fig. 3a–c). However, the overall topologies of the monomers in both structures (Fig. 2a, b) agree very well, as demonstrated also by superposition of the equivalent Cα atoms of both constructs yielding a root-mean-square deviation of only 0.6 Å (Fig. 3d). Also, structural details of the chromophore pocket are very similar, including the ZZEssa stereochemistry of BV and the exocyclic double bond of ring A of the chromophore, which results in sp3 hybridization of the C2 atom and therefore an out-of-plane oriented bond to the methyl group at this position. The chromophore interaction interface to the surrounding amino acids, especially potential hydrogen bonds, hydrophobic interactions and water molecules, are almost identical for the Pfr states of both Agp2-PCM variants (Fig. 2a, b; Supplementary Figs. 1, 2 and 3). In both Pfr structures and in contrast to other bathy phytochrome structures such as PaBphP-PCM[21], Gln190 of Agp2-PCM and Agp2-PAiRFP2 interacts with ring D of BV via a newly identified conserved water molecule (Fig. 2a, b;

Supplementary Fig. 3). This good structural agreement is mirrored by the far-reaching similarities of the RR spectra of the Pfr states of Agp2-PCM and Agp2-PAiRFP2 in solution as well as in the crystalline form (Supplementary Fig. 7), demonstrating that neither amino acid substitutions nor crystallization (e.g. crystal packing) do affect the structure of the chromophore pocket.

**Formation of the Meta-F state of Agp2-PAiRFP2.** Next, the Pfr crystals of Agp2-PAiRFP2 were irradiated with 785 nm light at ambient temperature to generate the intermediate state (Supplementary Fig. 8; Methods). In this case, irradiation for several seconds did not lead to non-diffracting crystals. The identity of the resultant state was determined by RR spectroscopy (Fig. 4). The RR spectra of Pfr measured from the crystal (trace **a**) and frozen solution (trace **b**) are nearly identical. Upon irradiation of Pfr at 130 K in frozen solution, Lumi-F is obtained (trace **c**), the spectrum of which displays the loss of the 811 cm⁻¹ hydrogen-out-of-plane (HOOP) band and upshift of the C=C stretching modes of the methine bridges (denoted as A-B, B-C and C-D)[28]. The subsequent intermediate Meta-F (trace **d**, frozen solution), formed upon irradiation at ca. 240 K, is reflected by further upshifts of the C-D and A-B stretching modes by ca. 10 cm⁻¹ and the drastic intensity loss of the B-C stretching, indicating the

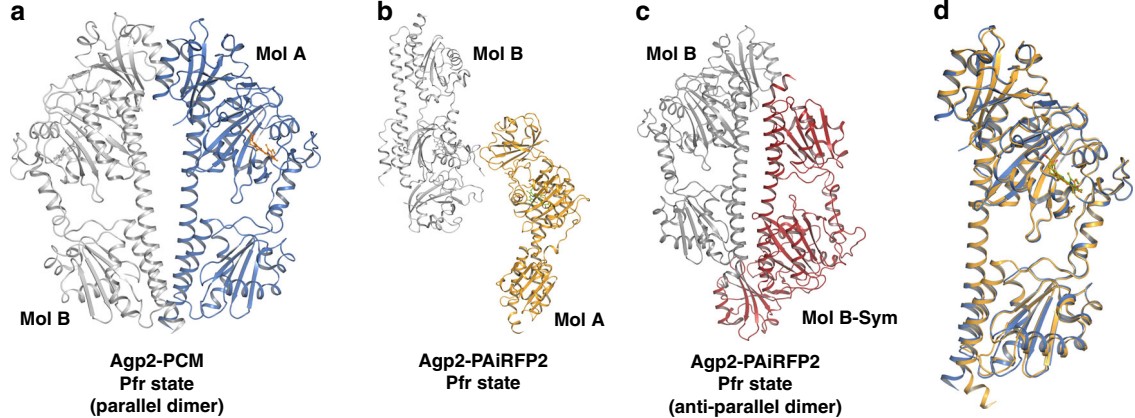

**Fig. 3** Dimer packing and superposition of wild-type Agp2-PCM and Agp2-PAiRFP2. In **a**, **b**, the crystallographically related monomers in the asymmetric unit are shown for **a** wild-type Agp2-PCM (parallel dimer) in blue and grey colour and **b** for Agp2-PAiRFP2 in orange and grey colour. **c** Symmetry-related crystallographic dimer (anti-parallel dimer) for Agp2-PAiRFP2 (Mol B - grey; Mol B-Sym - red). **d** A superimposition is shown for Mol A monomers of both crystal structures (wild-type Agp2-PCM - blue; Agp2-PAiRFP2 - orange), which reveal the high structural similarity of both crystal structures. All protein monomers are displayed as ribbons

relaxation of the ZZZssa chromophore structure. The spectrum of the irradiated Pfr crystals displays far-reaching similarities with that obtained from frozen solution irradiated under the same conditions and thus allows the unambiguous assignment to the Meta-F state. Minor differences exist between the RR spectra of the Meta-F states in frozen solution and in the crystalline form (trace **e**), such as the frequency upshift of the N-H in-plane (ip) mode of rings B and C. These deviations may point to a structural heterogeneity of the Meta-F state, presumably related to details of the hydrogen bonding network in the chromophore pocket. These findings point to the existence of Meta-F sub-states (vide infra). Note that, as for the solution spectra, the crystal spectra obtained after irradiation at 240 or 273 K were not different, implying that the chromophore adopts the fully relaxed structure of the final photoproduct already at 240 K (Supplementary Fig. 9).

The crystal structure of a Meta-F intermediate could be solved to a resolution of 2.16 Å (Table 1). The asymmetric unit includes two molecules (Mol A and B), which differ in terms of the intermolecular interactions of their N-termini (Fig. 5). Both show overall highly similar topologies. Compared to the Pfr structure, the overall fold of the protein and specifically the α-helical segment of the tongue remained unchanged but significant structural changes were observed for the chromophore and its binding pocket as reflected by the ΔFo (Fo$^{Meta-F}$−Fo$^{Pfr}$) electron density maps (Fig. 6). The C-D methine bridge has undergone the expected E→Z isomerization as also shown by 2$m$Fo-$D$Fc and several types of omit electron density maps (Supplementary Figs. 1c, f, i and 2c, f, i). The resulting ZZZssa stereochemistry of the chromophore structure exhibits a deviation of the tilt angle between neighbouring C and D pyrrole rings (38.9° and 44.5° for Mol A and Mol B, respectively; Supplementary Table 1). In fact, the RR spectra measured from five different crystals, grown and irradiated under similar conditions showed some spectral variations, specifically for the C-D methine bridge stretching mode, which varied in frequency between 1621 and 1624 cm$^{-1}$. The frequency of this mode sensitively responds to changes of the torsion of the C-D methine bridge[28], which is different for the Meta-F sub-states in Mol A and Mol B. It is therefore tempting to relate the abovementioned frequency changes to variable contributions of Mol A and B to the spectrum, which in turn may result from the preferential Raman enhancement of the two chromophores depending on the orientation of the crystal with respect to the laser beam.

The isomerization and the subsequent relaxation process of the chromophore are associated with structural changes of the protein that, for the sake of simplifying the discussion, may be sorted into three groups.

**Group I structural changes around the chromophore.** The first group includes a rearrangement of the extended hydrogen bond network surrounding the chromophore, most likely initiated by the almost 180° rotation of ring D (Figs. 2a–c, 6a, b; Supplementary Figs. 1–3, 6, 10) and facilitated by conformational changes of the participating amino acid residues and movements of ordered water molecules. The ring D N-H group, which in Pfr forms a hydrogen bond with the Asp196 side chain, now interacts with the structural water Wat795. The C=O group of ring D, originally in close proximity to Wat758 and its hydrogen bonded partner Gln190, is now placed in hydrogen bonding distance to His278, which is rotated and slightly shifted compared to the Pfr state. Concomitantly, the ring C propionate side chain of BV (prop-C) is transferred into a different hydrogen bonding environment. It involves Arg211, which has undergone major rotational motions towards prop-C, as well as His248, Ser260, Ser 262 and two water molecules (Wat707, Wat780) instead of Tyr165 and His278. The side chain of Phe244, which in wild-type Agp2 is a valine, has rotated to a position above the salt bridge between the ring B propionate side chain (prop-B) and Arg242, which together with Tyr205 has replaced Arg211 as the main hydrogen bonding partner (Fig. 2a, b, c; Supplementary Fig. 10). These changes are nearly identical for the Meta-F sub-states in Mol A and Mol B.

**Group II structural changes with switching amino acids.** The second group of structural changes in comparison to Pfr, as clearly seen in Mol A, involves Tyr165, Phe192 and Arg202, amino acid residues that are not connected to BV via a hydrogen bond network. These structural changes are accompanied by a transition from a β-facial to an α-facial orientation of the BV-binding Cys13 together with a conformational rearrangement of the N-terminus of the protein. In Mol A of the asymmetric unit, the N-terminus including Cys13 is completely free of intermolecular interactions within the crystal packing (Fig. 5). In the BV chromophore, the hydrophobic side of ring D, facing the tongue of the PHY domain, forms a hydrophobic pocket with the re-positioned Phe192 and Tyr165 (Figs. 6a, f, 7a, b, d, e), thereby

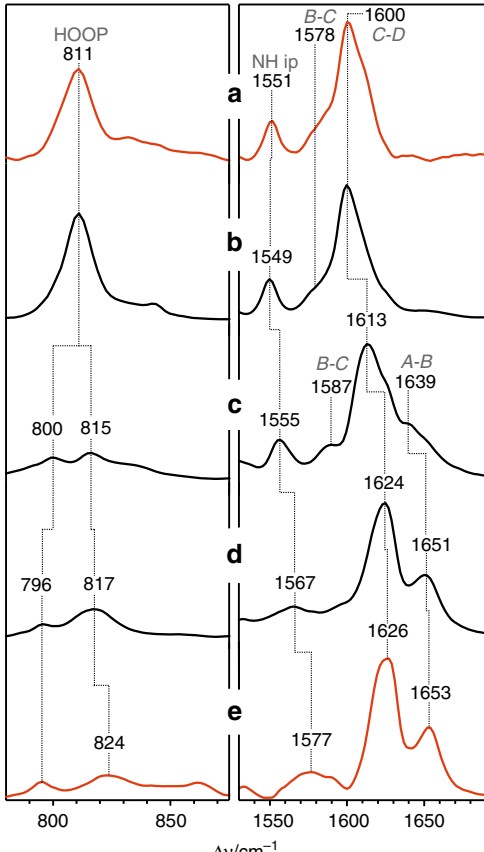

**Fig. 4** Experimental resonance Raman spectra of Agp2-PAiRFP2, measured from frozen solution and single crystals at 90 K. **a** Pfr, crystal; **b** Pfr, frozen solution; **c** Lumi-F, frozen solution obtained by irradiation at ca. 130 K; **d** Meta-F obtained by irradiation at ca. 240 K, frozen solution; **e** Meta-F obtained by irradiation at ca. 240 K, crystal. The RR spectra of Pfr measured from the crystal (trace **a**) and from frozen solution (trace **b**) are nearly identical. Upon irradiation of Pfr at 130 K, Lumi-F is obtained (trace **c**), whereas the subsequent intermediate Meta-F (trace **d**, frozen solution) was trapped upon irradiation at ca. 240 K. The spectrum of the irradiated crystals displays far-reaching similarities with that of Agp2-PAiRFP2 in frozen solution and irradiated under the same conditions and thus allows the unambiguous assignment of the Meta-F state (trace **e**). Note that, as for the solution spectra, there are essentially no spectral differences if the crystals are irradiated at 240 or 273 K. Only minor differences exist between the RR spectra of the Meta-F states in frozen solution and in the crystalline form (see also Supplementary Fig. 9)

replacing the conserved Wat758, accompanied by a positional shift of its interaction partner Gln190 away from ring D in the Meta-F state (Fig. 6a, g). Concomitantly, the side chain of Arg202 has rotated to accommodate two new water molecules (Wat722, Wat732) as a hydrogen bond bridge to Tyr165 (Fig. 6a, b, d). These conformational changes appear to be partially blocked in Mol B although its chromophore adopts nearly the same ZZZssa conformation and is involved in the same rearrangement of the hydrogen bond network as described as the first group of structural changes in Mol A (Fig. 7d, e). However, in Mol B, the side chains of both Tyr165 and Phe192 display two alternate conformations each, one conformation being similar to Pfr and the other resembling Mol A. In Mol B, only one partially occupied position of a water molecule (Wat809) is visible next to the rotated Tyr165 conformer and, moreover, Arg202 does not change its original position. In addition, Cys13 still adopts the original β-facial orientation of the Pfr state accompanied by an

unchanged N-terminus, which is tightly packed to Mol A (Fig. 5). In contrast, in the Pfr state of bathy phytochromes (e.g. wild-type Agp2-PCM, Agp2-PAiRFP2, *Pa*BphP-PCM (*P. aeruginosa*[29])) Tyr165 is in hydrogen bond distance to prop-C of BV, Phe192 points away from ring D of BV and Arg202 is facing towards BV (Fig. 7).

**Group III structural changes within the PHY tongue region.** The third group of structural changes refers to the PHY tongue (amino acids 433–471) that in Pfr adopts α-helical, loop and coiled structural regions but is converted to a β-sheet and a β-hairpin fold in Pr. In the structure of the Meta-F sub-states of Mol A and Mol B, these Pr characteristic features including the concomitant re-positioning of Trp440 away from the sterically pressing Gln190 were not visible. Instead, the amino acid segments 439–448 in Mol A and 439–442 in Mol B were not resolved in the electron density map, indicating a disordering of the polypeptide backbone. In Mol B, an additional hydrophobic contact in the dimer interface (Pro452[B]–Ser230[A]) is present and the segment is slightly better resolved than in Mol A (Fig. 5).

Structural changes in the tongue region are also reflected by the infrared (IR) difference spectrum of Agp2-PAiPFR2 in solution (Fig. 8, red trace), generated by subtracting the spectrum of Pfr (negative signals) from that of the final photoproduct (positive signals). In the amide I band region, no negative band at ca. 1657 cm$^{-1}$ is observed that in the Pr/Pfr difference spectrum of the full-length wild-type Agp2-FL (grey trace) demonstrates the restructuring of the α-helical part of the tongue segment of Pfr[17]. Instead, the negative signal at 1627 cm$^{-1}$ in the Agp2-PAiRFP2 difference spectrum (red trace) indicates the degradation of the coiled loop structure of the tongue segment, whereas the positive signal at 1642 cm$^{-1}$ points to the formation of a simpler β-hairpin segment. However, a more complex β-sheet formation can be ruled out due to the lack of a positive signal at ca. 1625 cm$^{-1}$. In the region >1680 cm$^{-1}$, the Agp2-PAiRFP2 difference spectrum agrees very well with that of the Meta-F/Pfr difference spectrum of wild-type Agp2-PCM (black trace), including the upshifts of the C=O stretching of the propionate side chain of ring C (from 1749 to 1761 cm$^{-1}$ for Agp2-PAiRFP2) and the C=O stretching of the ring D carbonyl from 1697 to 1709 cm$^{-1}$. In contrast, no notable signals are detected in the amide I band region of wild-type Agp2-PCM (black trace). Altogether, the IR spectra indicate the high structural similarity of the chromophore in the Meta-F states of Agp2-PAiRFP2 and wild-type Agp2-PCM, in line with the RR spectroscopic analysis.

However, unlike the wild-type Agp2-PCM in solution, the Meta-F states of Agp2-PAiRFP2 display the onset of the structural transformation of the tongue, documented by both the Meta-F crystal structure and the IR difference spectrum (after irradiation of 273 K in solution). The IR difference spectrum of irradiated Agp2-PAiRFP2 constitutes the final product of the phototransformation of Pfr. The only deviation between IR spectroscopic and crystallographic data of Agp2-PAiRFP2 refers to the formation of the small β-hairpin segment detected in the solution IR spectra, which may be impaired or not fully reached in the crystal. Nevertheless, the good overall agreement of the crystallographic and IR spectroscopic data also indicates that Agp2-PAiRFP2 in solution is not capable to perform the complete structural transformation of the tongue. To sum up, we assigned the crystallized intermediate to a precursor of Pr, i.e. more precisely to a Meta-F intermediate sub-state with a chromophore structure largely similar (albeit not identical) to the wild-type Meta-F state but including the onset of the restructuring of the tongue verified by our spectroscopic data.

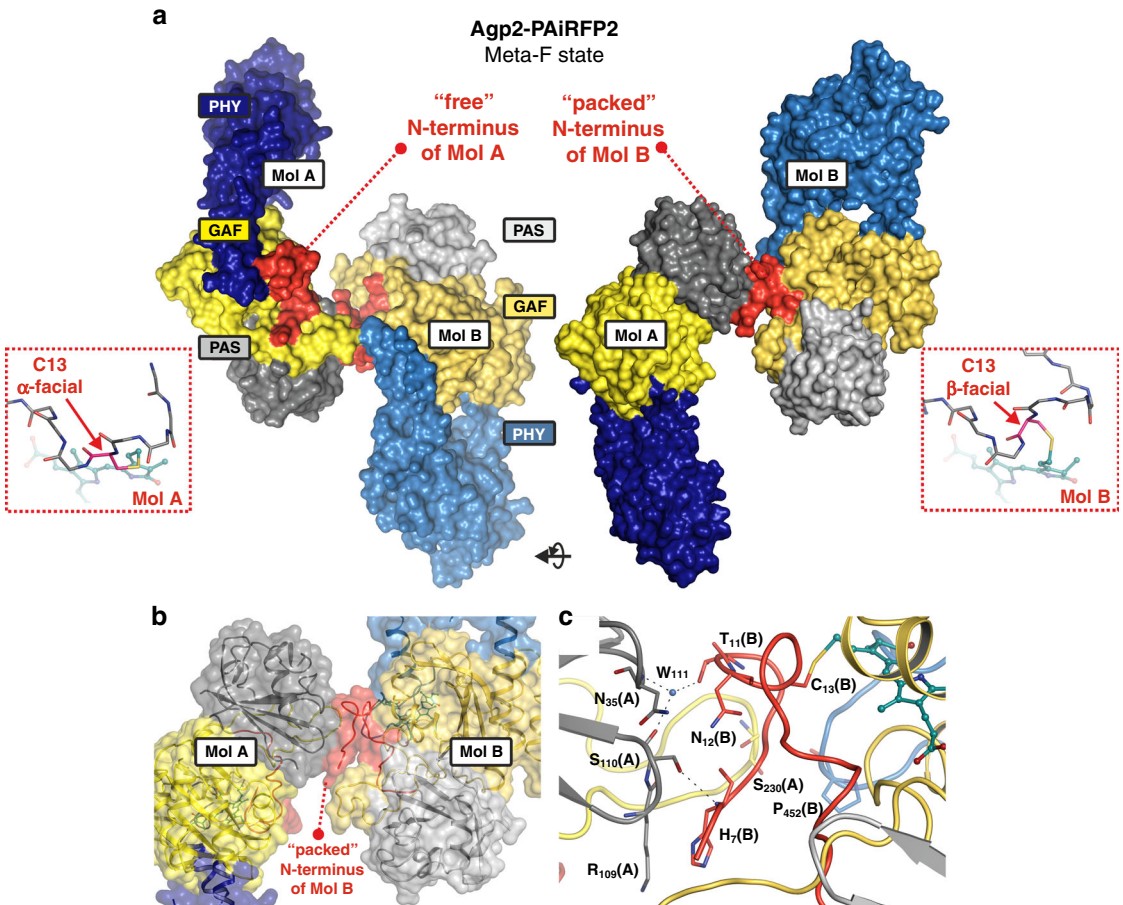

**Fig. 5** Crystal packing details of Agp2-PAiRFP2 monomers in the asymmetric unit. Surface representation of Agp2-PAiRFP2 (PDB entry 6G20) Mol A and B per asymmetric unit in the Meta-F sub-states. **a** (PAS, GAF, PHY in grey, yellow, blue, respectively; N-terminus highlighted in red) shows the free N-terminus for molecule A (Mol A), whereas the N-terminus of molecule B (Mol B) is packed against the PAS domain of Mol A. Owing to the higher flexibility, the N-terminus of Mol A can undergo a β-facial to α-facial orientation transition of the BV-binding Cys13 in the Meta-F state. **b** Close-up view into the packed N-terminal region of Mol B dimer shown as surface and cartoon representation. **c** Cartoon representation of the dimer interface of Mol A and B, including the N-terminus of Mol B with the β-facial orientation of Cys13. Amino acids involved in the interface are highlighted as sticks. Compared to Mol A (Fig. 7d, e), some amino acids in Mol B display a positional heterogeneity as reflected by the alternate conformations of Tyr165 and Phe192

## Discussion

In this work, we have presented the crystal structure of a Meta-F intermediate of a bathy phytochrome. This intermediate constitutes the starting point for the structural changes of the protein that eventually lead to the (de-)activation of the output module. The Meta-F state was generated from an Agp2 variant including 24 amino acid substitutions. Albeit mainly located outside the chromophore-binding pocket, the RR spectra of this intermediate marginally differs from that of Meta-F of the wild-type protein (Supplementary Fig. 9). Whereas spectroscopic data indicate nearly the same structure of the chromophore and its interactions with the immediate environment, Meta-F of Agp2-PAiRFP2 but not of the wild-type Agp2-PCM in solution displays already the onset of the restructuring of the tongue. This specific Meta-F state of Agp2-PAiRFP2, which cannot be easily trapped in the wild-type protein, provides valuable insight into the molecular mechanism by which chromophore isomerization is coupled to functional protein structural changes (Fig. 9). Furthermore, structural changes are more advanced in Mol A than in Mol B. Thus one may consider Mol B as a precursor of Mol A such that additional information about the sequence of the individual events can be derived. Accordingly, we propose the following mechanism for the Pfr-to-Pr photoconversion, including the Pr structure of Agp1 as a reference for the final state (Fig. 9).

(**1**) After illumination, the chromophore isomerizes from ZZEssa to ZZZssa stereochemistry, which had previously been observed in *Pa*BhpP for Lumi-F sub-states[21]. In contrast to Meta-F of Agp2-PAiRFP2, structural changes at the Lumi-F level are restricted to the chromophore and seem to propagate from the isomerization site at the C-D methine bridge to the B-C and eventually to the A-B methine bridge[21]. The primary consequence of ring D rotation is the replacement of Asp196 (Asp194 in *Pa*BphP) as the stabilizing group for ring D by His278 (His277 in *Pa*BphP). This interaction persists in Meta-F of Agp2-PAiRFP2 but in this state the chromophore has adopted a more relaxed structure with a nearly 180° rotation of ring D compared to Pfr (Fig. 9). Chromophore relaxation is accompanied by a substantial rearrangement of the hydrogen bond network involving prop-B and prop-C and the rotated ring D, which is observed in both Mol A and Mol B. These group I structural changes cause (**2**) reorientations of amino acids and segments in the environment of both ends of the chromophore (i.e. rings A and D), which only take place in Mol A. On the one hand, the side chains of Phe192 and Tyr165 are reoriented in such a way that their aromatic rings form a hydrophobic pocket with ring D (Fig. 9). On the other hand, the BV-binding Cys13 undergoes a change from a β-facial to an α-facial orientation together with a conformational rearrangement of the N-terminus of the protein

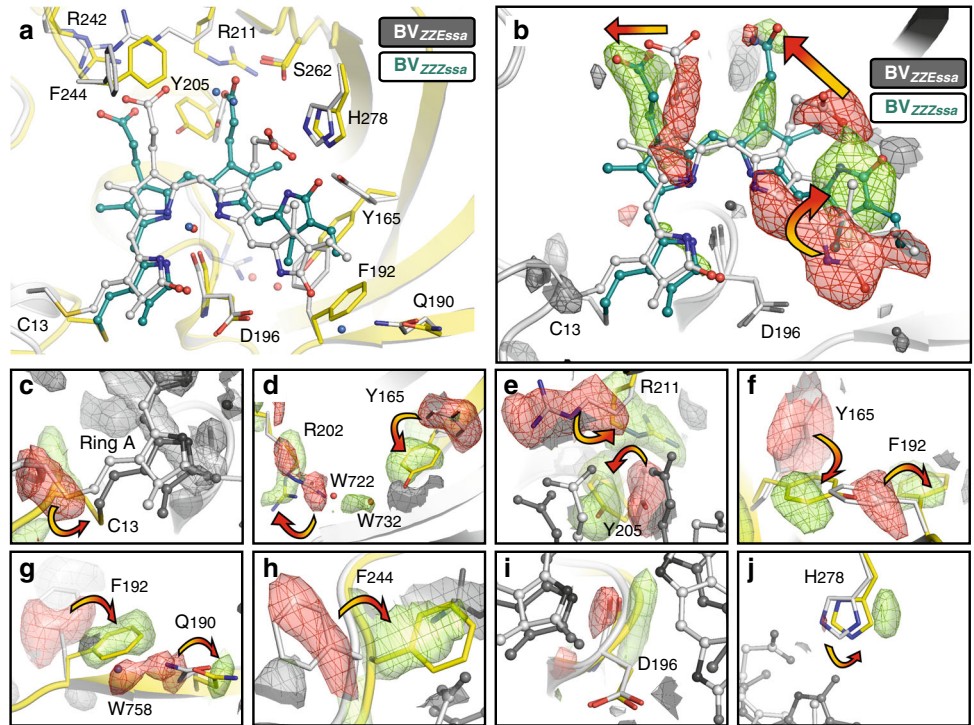

**Fig. 6** Superposition of Agp2-PAiRFP2 Meta-F and Pfr state. **a** Cartoon representation of the structures of the chromophore pocket. BV and amino acids of the chromophore pocket are shown as ball/sticks and sticks, respectively. In the Pfr structure, BV and amino acid side chains are coloured light grey and water molecules are depicted as blue spheres. In the Meta-F sub-state structure (Mol A), BV is coloured teal, protein in yellow and water molecules are depicted as red spheres. **b** The Fo$^{Meta-F}$−Fo$^{Pfr}$ electron density map of BV shows structural changes in the chromophore associated with the Pfr→Meta-F transition. Expanded views of sections of **b** are shown in **c–j**. Positive Fo$^{Meta-F}$−Fo$^{Pfr}$ electron density map is coloured green, whereas negative ΔFo electron density is coloured in red. For the sake of clarity, ΔFo electron density and amino acids that do not belong to the highlighted amino acid in the corresponding figure were coloured light and dark grey. In **c–j**, ΔFo electron density is contoured at 3.0 σ and in **b** at 4.0 σ

(Figs. 6c, 7d, e). Since the latter is impaired in Mol B most likely due to the tight packing of the N-terminus (Fig. 5), it is tempting to assume that the structural rearrangements in the vicinity of rings A and D may represent a concerted process that can only proceed in Mol A (Fig. 7e, e). (3) Owing to the changes around ring D, the water molecule Wat758 (Fig. 9, red sphere) at ring D is displaced, accompanied by (4) a positional shift of Gln190 in the Meta-F intermediate, which has a direct impact on the PHY tongue. In the Pfr state, the highly conserved Trp440 of the PHY tongue overlaps with the position of this shifted Gln190 (Meta-F state). (5) Therefore, the movements of Gln190 and Trp440 eventually affect the PHY tongue (Agp2-PAiRFP2) in Meta-F, which conceivably leads to refolding of the tongue also indicated by the distant positions of Trp440 (Trp445 in Agp1) and the Gln190 (Leu191 in Agp1) homologues in the refolded part in the Pr state of Agp1-PCM$_{SER13}$[30]. In Agp2-PAiRFP2 (Fig. 9c, left panel), Trp440 is part of the tongue segment that is disordered in Agp2-PAiRFP2 Meta-F (middle panel). For comparison Agp1-PCM$_{SER13}$ shows an ordered segment and a β-sheet in this region (right panel). In this sense, the Gln190-Trp440 couple may represent a trigger for the reorganization of the tongue (group III structural changes), which takes place in at least two steps, starting with the reorganization of the loop-coil region as observed in Mol A of the Meta-F state (Agp2-PAiRFP2) that precedes the transition of the α-helical segment to a β-sheet structure.

Now we ask for the generality of the proposed mechanism. The group II structural changes seem to constitute the key process for coupling chromophore and protein structural changes. Here the formation of the hydrophobic pocket for ring D via reorientation

of Tyr165 and Phe192 and the hydrogen bonding interactions of Tyr165 with the rotated Arg202 via two new water molecules is a structural motif that can be deduced from the various Pr state structures of prototypical bacterial (Agp1-PCM$_{SER13}$ of *A. fabrum* [PDB entry 5HSQ[30]], *Dr*BphP-PCM of *D. radiodurans* [e.g. PDB entry 4Q0J[18]]) and cyanobacterial (Cph1-PCM of *Synechocystis* sp. [PDB entry 2VEA[31]]) phytochromes. Also in plant phytochrome *At*PhyB-PCM of *A. thaliana* (PDB entry 4OUR[32]) in its (dark adapted) Pr state, very similar rotameric states of the corresponding amino acids were found although the two water molecules between the Tyr and Arg residues were not resolved in that structure due to the low resolution (3.4 Å) (Fig. 7f–i).

In the Pfr structures of the bathy phytochrome *Pa*BphP and the prototypical phytochrome of *Dr*BphP, the conformations of all three amino acids (Tyr165, Phe192 and Arg202 in Agp2) are very similar to those observed in the present Pfr structures of Agp2-PCM and Agp2-PAiRFP2. Consistent with these results, alternating conformations of Tyr165, Phe192 and Arg211 (numbering of Agp2) have been observed in the mixed Pr/Pfr state crystals of the Q188L *Pa*BphP variant[29]. Moreover, the reorientation of the chromophore-binding Cys is not restricted to Agp2 since all known structures show an α-facial and β-facial orientation of the chromophore-binding Cys in Pr and Pfr, respectively (Fig. 7; Supplementary Fig. 11). In view of these findings, we hypothesize that the group II structural changes are a common mechanistic element for the Pfr→Pr phototransformations of all phytochromes.

The group III structural changes refer to the structural transition of the tongue and its initiation by Gln190/Trp440. Here we note similarities in the structural data for the prototypical

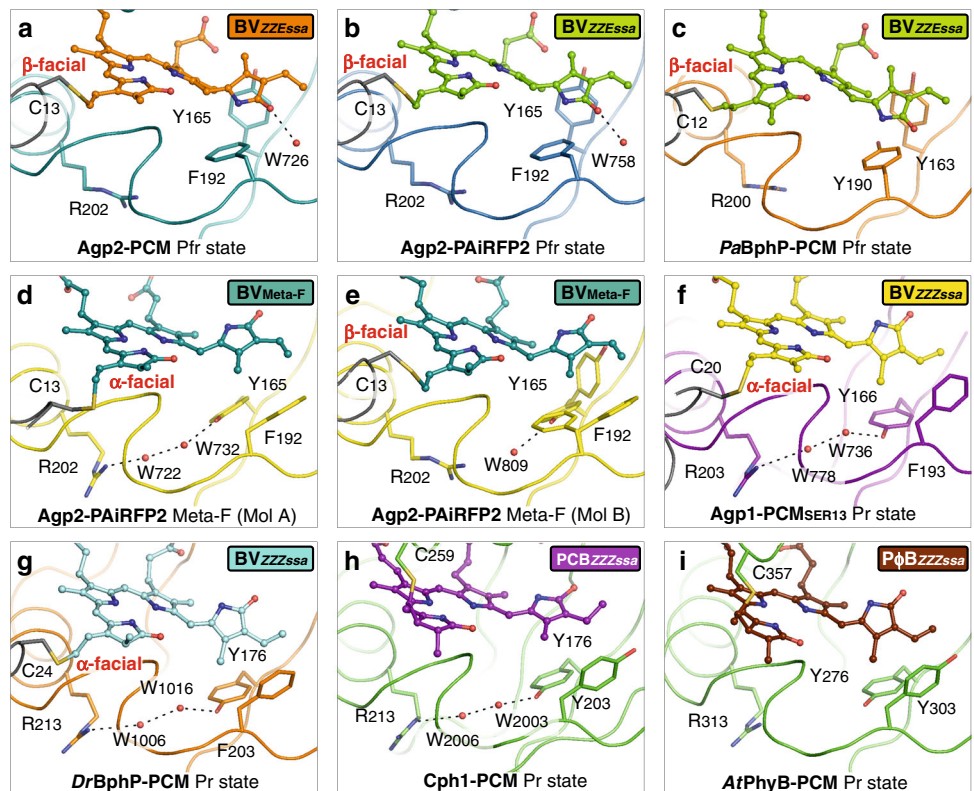

**Fig. 7** Structural comparison of various phytochromes in different states of the photocycle. Proteins in ribbon representations, bilin chromophores in ball/stick representations. Selected conserved amino acids are highlighted as sticks, water molecules are shown as red spheres. In the Pfr state of bathy bacteriophytochromes (**a**, wild-type Agp2-PCM and **b**, Agp2-PAiRFP2; **c**, *Pa*BphP-PCM (*P. aeruginosa*; PDB entry 3NHQ[29])) Tyr165, Phe192 and Arg202 show a conserved conformation. Furthermore, the cysteines are always β-facially bound to BV. In the Meta-F state of Agp2-PAiRFP2, the two monomers (Mol A and B) of the asymmetric unit (**d**, **e**) show slightly different structures. In Mol A, the N-terminus including Cys13 is free of intermolecular interactions (Fig. 5). After illumination, Mol A reveals the BV isomerization and Cys13 in α-facially bound position accompanied by a conformational shift of the N-terminus and a conformational change of Phe192, Tyr165 and Arg202, together with the formation of a hydrogen bond bridge mediated by two water molecules (**d**, Wat722 and Wat732) between Tyr165 and Arg202. In Mol B (**e**), the N-terminus is tightly packed to Mol A (Fig. 5). In the Pr states of prototypical bacterial (**f**, Agp1-PCM_SER13 of *A. fabrum*, PDB entry 5HSQ[30] and **g**, *Dr*BphP-PCM of *D. radiodurans*; e.g. PDB entry 4Q0J[18]), cyanobacterial (**h**, Cph1-PCM of *Synchocystis* sp.; PDB entry 2VEA[31]) and plant (**i**, *At*PhyB-PCM of *A. thaliana*; PDB entry 4OUR[32]) phytochromes, the rotameric states of the corresponding amino acids as well as the same two water molecules between the Tyr and Arg residues are similar to Mol A in the Meta-F state of Agp2-PAiRFP2 (**d**). Because of the low resolution (3.4 Å), these water molecules were not resolved in the *At*PhyB-PCM structure[32]

*Dr*BphP[18] since His201 undergoes a comparable conformational change as its counterpart Gln190 in Agp2. Trp451 of *Dr*BphP (Trp440 in Agp2), which is resolved in three of the four molecules within the asymmetric unit of the Pfr-like structure, shows different conformations, which one may take as an indication for an increased flexibility at this position. Additionally, the tongue region is also not completely resolved. Highly flexible regions (unresolved regions in *Dr*BphP Mol A 454–459, Mol B 452–459, Mol C 450–462, Mol D 452–463) comparable with those in the present Meta-F sub-state structures (Mol A 439–448, Mol B 439–442) support the view that restructuring of the tongue region starts (Pfr→Pr photoconversion) in this region and subsequently propagates to the rest of the tongue.

After the secondary structure transition of the tongue is completed, the domain arrangement within the PCM is changed as well[18–20]. These major conformational changes in the protein, which occur during the transfer from the Meta intermediate to the final product of the phototransformation, were proposed to affect the structure and thus the activity status of the output module. The present results, however, do not provide any information about these final steps of the photo-induced reaction cascade and the role of the output module in signalling, folding and back reaction as shown by other experiments (e.g. refs. [8,32–35]). This is also true for the possible

role of dimerization. In this context, we wish to point out that the organization of monomers within the crystal packing do not allow for any conclusions about functionally relevant and concentration-dependent protein–protein interactions in solution, specifically when, as in the present case, surface residues have been substituted (e.g. ref. [30]).

Taken together, we have presented the crystal structure of an intermediate that is placed in a key position for coupling the photoconversion of the photoreceptor module with the (de-)activation of the output module. In view of similar structural motifs, the proposed general mechanistic model for the Pfr→Pr photoconversion of the bathy phytochrome Agp2, as summarized in Fig. 9, may also hold for prototypical phytochromes, in line with previous suggestions for the Pr→Pfr photoconversion[18]. With the structural and spectroscopic data presented here, we are getting closer to identifying the mechanistic features that are common to the photocycles in both prototypical and bathy phytochromes towards a general scheme with small subtype-specific differences.

## Methods

**Molecular cloning of wild-type Agp2-PCM**. The Agp2-PCM gene (NCBI GenBank ID AAK87910) was PCR-amplified from *A. fabrum* genomic DNA and cloned into a pET21b expression vector with C-terminal His-tag by using the

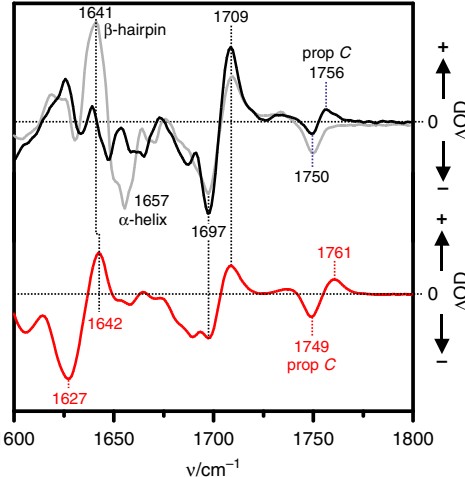

**Fig. 8** Photoinduced infra-red (IR) difference spectra of Agp2 variants in solution. Black trace (top): wild-type Agp2-PCM difference spectrum obtained after irradiation at ca. 240 K corresponding to the Meta-F-minus-Pfr difference spectrum (taken from ref. [17]); red trace (bottom) Agp2-PAiRFP2, difference spectrum obtained after irradiation at 273 K. The grey trace (top) refers to the difference spectrum Pr-minus-Pfr of full-length wild-type Agp2, generated upon irradiation at 273 K (spectrum taken from ref. [17]). Note that the Pr state could not be enriched for the wild-type Agp2-PCM due to the fast thermal back conversion to the Pfr state[60]. In each spectrum, the negative signals refer to Pfr, whereas the positive signals correspond to the respective photoproduct

following primers: forward primer sequence ATGTATATCTCCTTCTTA AAGTTAAAC and reverse primer sequence CATCACCATCACCATCACTAA GATCCG (full DNA sequence in Supplementary Table 2)[36,37]. The gene encoding the PCM module derived from Agp2 (1–501 amino acids plus hexa-histidine tag) was transformed into *Escherichia coli* BL21-DE3 cells (Agilent Technologies).

**Molecular cloning of Agp2-PAiRFP2.** The DNA sequence of Agp2-PAiRFP2 (NCBI GenBank ID AGS83373.1)[25] was codon optimized (Supplementary Table 2)[38]. The gene was synthesized by GENEWIZ Inc. and cloned into a pET21b expression vector with C-terminal His-tag and transformed into *E. coli* BL21-DE3. The construct contains the following 24 substitutions: Lys69Arg, Arg83Lys, Gly120Asp, Ala123Thr, Met163Leu, Gln168Glu, Arg220Pro, Ser243-Asn, Val244Phe, Gly269Asp, Ala276Val, Tyr280Cys, Glu294Ala, His303Phe, His333Arg, Ile336Leu, Asp349Arg, Met351Ile, Ala386Val, Gly409Asp, Leu419Ile, Thr469Ser, Ala487Thr, Glu494Gly (Supplementary Figs. 4, 12).

**Purification of wild-type Agp2-PCM and Agp2-PAiRFP2.** Agp2-PCM and Agp2-PAiRFP2 constructs were expressed using an autoinduction medium (Overnight Express™ Instant TB Medium; Novagen) for 48 h and 20 °C. Cell pellets were washed and cell lysis was carried out using cell fluidizer (Microfluidics, Newton USA) in 50 mM Tris-HCl buffer containing 50 mM sodium chloride at pH 7.8, 5% glycerol, 2 mg mL⁻¹ lysozyme (Merck Millipore), 60 µg mL⁻¹ DNAse (Sigma-Aldrich), 1 mM MgCl₂ and 0.5 mM phenylmethanesulfonyl fluoride (Sigma-Aldrich). Lysed cells were centrifuged and protein in the supernatant was precipitated with 2 M ammonium sulphate. The pellet was dissolved with 50 mM Tris/HCl, 10 mM imidazole, 400 mM sodium chloride at pH 7.8 and loaded with the same buffer on a Ni-NTA column (5 mL HP columns; GE Healthcare). Purified apo-phytochrome was eluted with a linear imidazole gradient. Imidazole was removed by ammonium sulphate precipitation. The chromophore BV (Frontier Scientific) was added at ~3× molar excess to the protein. Holo-protein was concentrated by ammonium sulphate precipitation and redissolved in size exclusion buffer (20 mM HEPES buffer pH 7.5 containing 150 mM sodium chloride). Size exclusion chromatography (HiLoad Superdex 200 column, GE Healthcare) was performed and yielded pure protein at a concentration of 30 mg mL⁻¹.

**Methylation of wild-type Agp2-PCM.** Agp2-PCM was methylated at a concentration of 15 mg mL⁻¹ in 20 mM HEPES buffer, 150 mM sodium chloride, 1 mM tris(2-carboxyethyl)phosphine (TCEP) and 1 mM EDTA at pH 7.7 using a JBS Methylation Kit (Jena Biosciences) according to the manufacturer's protocol. Subsequently, the protein was washed 3× using concentrators (10 kDa cutoff;

Millipore concentrator) in the same buffer. The methylated sample was recovered with negligible losses and concentrated to 30 mg mL⁻¹.

**Crystallization of wild-type Agp2-PCM.** Methylated Agp2-PCM crystallized at a concentration of 30 mg mL⁻¹ in 30–40% MPD, 5–15% PEG 8000 and MES (pH range 5.5–9) at 279 K using the vapour diffusion method in linbro 24-well plates. First crystals appeared after 1 week reaching their final size after 3–4 weeks with dimensions 0.2 × 0.2 × 0.8 mm³. Crystals were immediately frozen in liquid nitrogen.

**Crystallization of Agp2-PAiRFP2.** Agp2-PAiRFP2 was crystallized at a concentration of 25 mg mL⁻¹ in darkness at 279 K within a precipitant solution containing 1.0–2.2 M ammonium sulphate, 2–12% PEG 1000 and 0.1 M HEPES pH 6.8–7.7. First crystals appeared after 1 week reaching their final size after 3–4 weeks with dimensions 0.03 × 0.03 × 0.1 mm³. Crystals were then transferred under safe light to a cryo-protectant consisting of 20% glycerol in crystallization buffer and immediately frozen in liquid nitrogen. For photoconversion experiments, Agp2-PAiRFP2 crystals were irradiated at 279 K with a 785 nm cw-laser (100 mW, diode laser, Minasales, Germany) in the crystallization drop for different times. For each illumination time (1 s, 2 s, 3 s (finally used to trap Meta-F), 4 s, 5 s, 7 s, 11 s, 15 s), a synchrotron data set at 100 K could be collected at ESRF (Grenoble, France). Simultaneously, crystals from the same crystallization drop were used for RR spectroscopic measurements. Alternatively crystals were frozen under safe light and illuminated from different sides at the beamline while cryo-cooling was switched off for defined periods of time.

**Data collection and structure analysis.** Diffraction data sets of >100 crystals were obtained at 100 K using synchrotron X-ray sources at ESRF (Grenoble, France) and BESSYII (Berlin, Germany). The best diffraction data for the highest resolution of wild-type Agp2-PCM Pfr state were collected under green safe light at the synchrotron ESRF (Grenoble, France) beamline ID30A-3[39] with a Pilatus3_2M detector at λ = 0.968 Å.

The best diffraction data for the highest resolution of dark adapted Agp2-PAiRFP2 Pfr state were collected at the synchrotron BESSYII (Berlin, Germany) beamline BL14.1[40] with a Pilatus 6 M detector at λ = 0.918 Å. Diffraction data for the illuminated state were collected at ESRF (ID23–1[39]) with a Pilatus 6 M detector at λ = 0.972 Å. All images were indexed, integrated and scaled using the XDS program package[41] and the CCP4[42] programs SCALA[43] and AIMLESS[44]. All wild-type Agp2-PCM crystals belong to the orthorhombic space group P2₁2₁2₁ (for wild-type Agp2-PCM: a = 74.5 Å, b = 93.4 Å, c = 174.4 Å, α = β = γ = 90°). All Agp2-PAiRFP2 crystals belong to the hexagonal space group P6₃22 (for Agp2-PAiRFP2: a = b = 182 Å, c = 180 Å, α = β = 90°, γ = 120°). Table 1 summarizes the statistics for crystallographic data collection and structural refinement.

Initial phases for Agp2-PCM and Agp2-PAiRFP2 were obtained by the conventional molecular replacement protocol (rotation, translation, rigid-body fitting) using the *Pa*BphP-PCM (PDB entry 3C2W) and Agp1-PCM structures (PDB entry 5HSQ) as initial search models, respectively[27,30]. After excluding BV from the initial search models, molecular replacement was used in the program Phaser[45]. PAS, GAF and PHY domains were placed separately in the molecular replacement search. Simulated annealing with the resulting model was performed using a slow-cooling protocol and a maximum likelihood target function, energy minimization and B-factor refinement by the program PHENIX[46]. Crystallographic refinement was carried out in the resolution range of 45.09–2.50 Å for wild-type Agp2-PCM in its Pfr state, 47.73–2.03 Å for the Agp2-PAiRFP2 in its Pfr state and 47.79–2.16 Å for Agp2-PAiRFP2 in its Meta-F state. After the first rounds of refinement for all three data sets, the BV chromophore in the ligand-binding pocket was clearly visible in the electron density of both σ_A-weighted 2mFo-DFc maps, as well as in the σ_A-weighted simulated annealing omitted density maps (Supplementary Figs. 1, 2).

All crystallographic structures were modelled with TLS refinement[47] using anisotropic temperature factors for all protein atoms. Restrained, individual B-factors were refined, and the crystal structures were finalized by the CCP4 program REFMAC5[48] and other programs of the CCP4 suite[42]. The final model has agreement factors $R_{free}$ and $R_{cryst}$ of 24.1 and 20.1% for wild-type Agp2-PCM in its Pfr state (PDB entry 6G1Y), of 21.3 and 18.0% for Agp2-PAiRFP2 in its Pfr state (PDB entry 6G1Z) and of 23.2 and 20.9% for Agp2-PAiRFP2 in its Meta-F state (PDB entry 6G20), respectively. Manual rebuilding of the crystal structure models and electron density interpretation were performed after each refinement step using the program COOT[49]. Structure validation was performed with the programs PHENIX[46], SFCHECK[50], WHAT_CHECK[51], MolProbity[52] and RAMPAGE[53]. Potential hydrogen bonds and van der Waals contacts were analysed using the programs HBPLUS[54] and LIGPLOT 1.45+[55]. All ΔFo electron density maps were calculated with the PHENIX program suite[46]. All other electron density maps were calculated with the CCP4 program FFT[56]. All crystal structure superpositions of backbone α-carbon traces were performed using the CCP4 program LSQKAB[42]. All molecular graphics representations in this work were created using PyMol[57].

**RR and IR spectroscopy.** RR measurements were performed using a Bruker Fourier-transform (FT) Raman spectrometer RFS 100/S with 1064 nm excitation

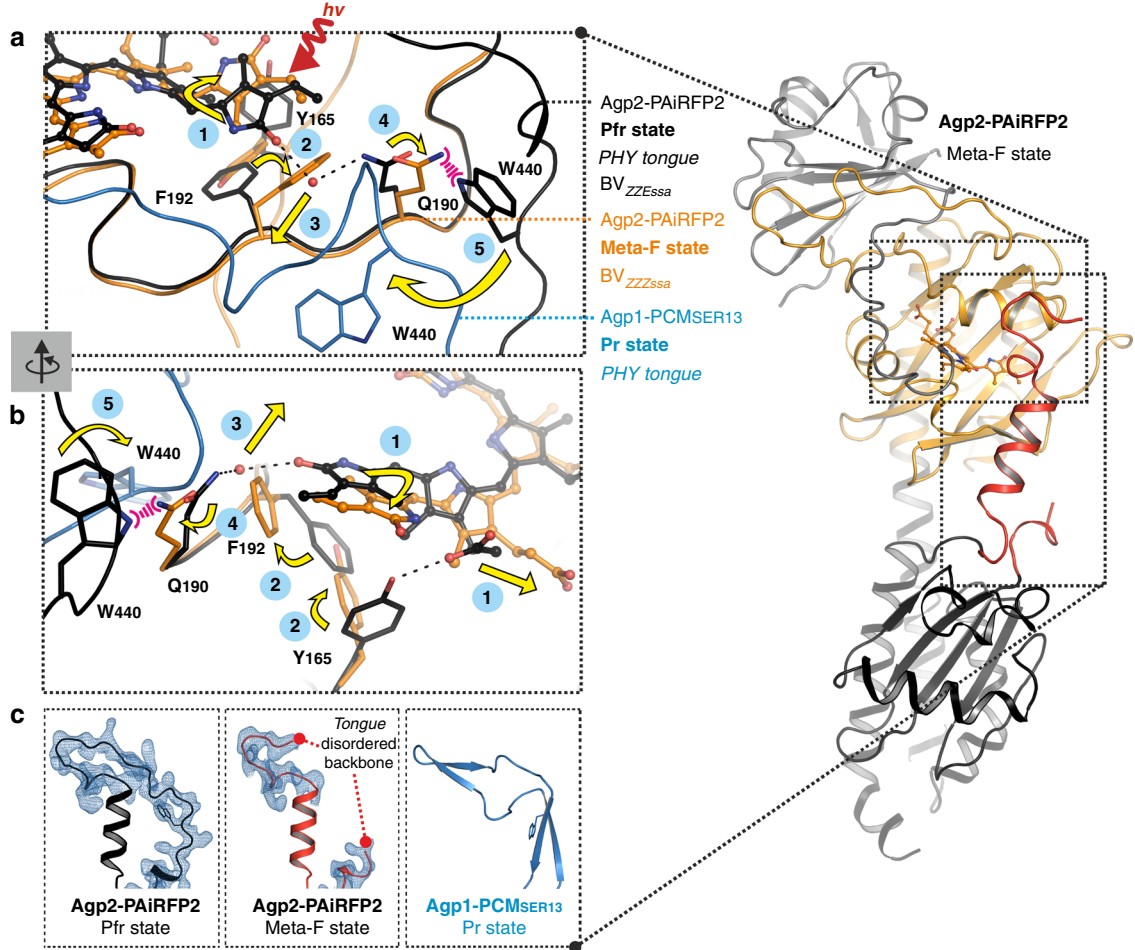

**Fig. 9** Proposed structural mechanism for the sequence of events during Pfr-to-Pr photoconversion. The upper and middle left panels (**a**, **b**) show the relevant section of a superposition of the structures of Pfr (Agp2-PAiRFP2, PDB entry 6G1Z, black), Meta-F (Agp2-PAiRFP2, PDB entry 6G20 (Mol A), orange, PHY tongue in red) and Pr (Agp1-PCM$_{SER13}$, PDB entry 5HSQ[30], blue) for comparison. Amino acid residues Tyr165, Gln190, Phe192 and Trp440 (Agp2-PAiRFP2 numbering) are shown as sticks, BV is depicted as balls and sticks. Following (**1**) E→Z photoisomerization of BV (Pfr→Lumi-F) and the subsequent chromophore relaxation, the rearrangement of the hydrogen bond network and the rotated ring D induces (**2**) the reorientation of the side chains of Phe192 and Tyr165 such that their aromatic rings form a hydrophobic pocket with ring D. Thus (**3**) the water molecule Wat758 (red sphere) at ring D is displaced, accompanied by (**4**) a positional shift of Gln190 in the Meta-F intermediate, which has a direct impact on the PHY tongue. In the Pfr state, the highly conserved Trp440 of the PHY tongue overlaps with the position of this shifted Gln190 (Meta-F state). Therefore, steric hindrance by Gln190 (**5**) would cause Trp440 to move and eventually affect the PHY tongue (Agp2-PAiRFP2) in Meta-F to induce refolding upon transition to the Pr state. **c** In Agp2-PAiRFP2 (left panel), Trp440 is part of the tongue segment that is disordered in Agp2-PAiRFP2 Meta-F (middle panel). For comparison, Agp1-PCM$_{SER13}$ shows an ordered segment and a β-sheet in this region (right panel)

(Nd-YAG cw laser, line width 1 cm$^{-1}$), equipped with a nitrogen-cooled cryostat from Resultec (Linkam). All spectra of the samples in frozen solution were recorded at 90 K with a laser power at the sample of 780 mW with a typical accumulation time of 1 h. In order to identify potential laser-induced damage of the phytochrome samples, spectra before and after a series of measurements were compared. In no case, changes between these control spectra were determined. Protein and buffer Raman bands were subtracted on the basis of a Raman spectrum of apo-phytochrome. The intermediate states Lumi-F and Meta-F were accumulated by 785 nm irradiation of the Pfr sample at 133 and 243 K, respectively. Residual contributions of non-photolysed Pfr to the RR spectrum measured at 90 K were subtracted. The spectra of the Pr state were obtained as described elsewhere[17,58]. RR measurements of single protein crystals were performed using a Ramanscope III Raman microscope connected to the RFS-100 FT-Raman spectrometer (Bruker Optics) via an optical fibre. The laser power of the 1064 nm excitation line was set to 200 mW at the sample. Protein crystals of the dark (Pfr) or photoinduced states (Meta-F) were measured at 90 K[59].

For IR spectroscopic measurements, the protein samples were placed between two BaF$_2$ (CaF$_2$) windows (15/20 mm diameter) with a 3-μm-thick PTFE spacer and equilibrated at 273 K. IR spectra were recorded in a Bruker IFS66v/s or IFS28 spectrometer equipped with a liquid nitrogen cooled MCT detector (J15D series, EG&G Judson). Spectra were recorded 2 min prior to illumination and 30 s after the illumination and subsequently the Pr-minus-Pfr difference spectra were calculated by subtracting the Pfr spectra from the Pr spectra.

## Data availability

The atomic coordinates and structure factors have been deposited in the Protein Data Bank under accession codes 6G1Y, 6G1Z and 6G20. A reporting summary for this article is available as a Supplementary Information file. Other data are available from the corresponding authors upon reasonable request.

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

## Acknowledgements

We thank Anja Koch and Brian Bauer for assistance in purifying reagents. We are grateful to Manfred Weiss and the scientific staff of the BESSY-MX/Helmholtz Zentrum Berlin für Materialien und Energie at beamlines BL14.1, BL14.2 and BL14.3 operated by the Joint Berlin MX-Laboratory at the BESSY II electron storage ring (Berlin-Adlershof, Germany) and the scientific staff of the European Synchrotron Radiation Facility (ESRF, Grenoble) at beamlines ID23-1, ID23-2, ID29, ID30A-1, ID30A-3 and ID30B for continuous support. This work was supported by grants from the Deutsche Forschungsgemeinschaft (SFB1078-B6 to P.H. and P.S.; SFB1078-C3 to M.A.M.; SFB740-B6 to P.S.), Integrated Graduate School fellowship of SFB1078 to L.S., European Synchrotron Radiation Facility (ESRF beamtime BAG to N.K., D.v.S. and P.S.), DFG Cluster of Excellence 'Unifying Concepts in Catalysis' (Research Field E4 to M.A.M., P.H. and P.S.) and the Einstein Center of Catalysis EC$^2$ Berlin (to P.H., F.V.E., M.A.M. and P.S.).

## Author contributions

P.S. and P.H. conceived, planned and designed the project; P.S. coordinated the project; L.S., B.M.Q. and N.M. performed wild-type Agp2-PCM and Agp2-PAiRFP2 preparation and functional analysis, B.M.Q., A.S., T.S. and P.S. performed wild-type Agp2-PCM crystallization and crystal optimizations; A.S., L.S. and M.S. performed Agp2-PAiRFP2 crystallization and crystal optimizations, L.S. and N.M. performed UV-vis spectroscopy on samples; A.S., L.S., M.S., B.M.Q., T.S., D.K., D.v.S. and P.S. performed X-ray diffraction data collections; M.F.L., F.V.E. and D.B. performed resonance Raman (RR) and infra-red spectroscopy on solution and crystal samples, M.A.M. and P.H. supervised Raman spectroscopic experiments; A.S., L.S. and P.S. performed structural analysis; A.S., L.S. and P.S. prepared all structure figures; T.L. provided phytochrome alignments; all authors contributed to data analysis and interpretation; A.S., L.S., N.K., T.L., P.H. and P.S. wrote the paper.

## Additional information

**Competing interests:** The authors declare no competing interests.

