## [Peer Review File · Nature Communications]

Reviewers' comments:

Reviewer #1 (Remarks to the Author):

The manuscript by Schmidt et al. reports crystallographic snapshots of the bathphytochrome Agp2 in its Pfr ground state as well as of (a) structural intermediate(s) along the far-red light induced conversion to the Pr state. The authors employ a combination of X-ray crystallography and spectroscopy techniques to assign this intermediate as the meta-F intermediate implicated in the conversion from Pfr to Pr. Schmidt et al. observe a set of structural rearrangements that are comparable to changes seen in canonical bacteriophytochromes featuring a Pr ground state. Eventually, the authors attempt to discuss the implications of the observed restructuring in the context of the signal transduction pathways of phytochromes in general.

This study is very well performed and the authors present their results in a conclusive way. However, the short main text passage creates the impression that especially the implications of this work in the context of the phytochrome field are treated rather superficially. Also some of the conclusions from experimental data are presented in the figure legends rather than the main text, making it difficult to read the manuscript in a coherent manner. To improve the readability of the manuscript and to rule out potential inconsistencies/ambiguities in data presentation, I would recommend to work on the following points:

1 - The authors report structures of Agp2-wt and of the PAiRFP2 variant. It is mentioned that the two proteins crystallize in different space groups and also show different oligomeric assemblies, but no visual comparison is provided. Especially since the (parallel) dimeric assembly is implicated to be important for long range signal transduction, it is puzzling that no structure of the parallel wt-Agp2 dimer is shown and consequently discussed in the context of proposed signal transduction pathways.

2 - Along the line of the previous argument, the authors state in their summary that "molecular mechanisms of sensor-output communication remain elusive due to the lack of structural information of meta intermediates". This appears overstated, because the ultimate coupling with the output is achieved via the different Pr and Pfr environments. Structural hypothesis for how these different environments are linked to the output have been put forward for the *Deinococcus*, *Xanthomonas*, *Idiomarina* and *Rhodospseudomonas* phytochrome systems where structural information for domains following the PCM is available. These systems and sensor-output communication pathways are not discussed at all. Nevertheless, structural information of meta intermediates would be highly appreciated to better understand the initial rearrangements that precede restructuring of the tongue element and how these are potentially linked to tongue reordering.

3 - In order to obtain the major results of this study, a variant with 24 substitutions is used. The authors state that all of these amino acid replacements do not affect essential structural and mechanistic properties. I would assume that the authors have tried to crystallize the variant in the same conditions as the wild-type (and vice versa). If this has not worked, then the lack of parallel dimerization of the PAiRFP2 variant already suggests some important structural consequences. Some information with respect to the in solution oligomeric state of both proteins could be helpful to appreciate the different crystallization behavior.

Also the fact that only one substitution is close to the BV environment is not correct - also M163L and A276V are positioned close to the D-ring, which is part of the central structural rearrangements. Similarly, S243N could impact the BV environment and the light induced structural rearrangements.

The most important point in this context is the observed difference with respect to the stability of illuminated crystals (wild-type crystals dissolve). The presentation of data should be more complete to enable an appreciation of the involvement of crystal lattice restraints. The contacts of the N-termini are shown and discussed but there must be more crystal contacts that stabilize the

assemblies. Are there differences in the tongue environments? It is stated that the wt structure does not have any contact in the tongue region but this is not explicitly stated for the variant. Could different contacts in the tongue region (in addition to the contacts of the N-termini) also be the reason for the different observations in mol A and mol B upon illumination?

4 - The central spectroscopic data to assign the meta-F intermediate are mentioned rather superficially in the main text. Most of the results and discussion are presented in Figure legends (Supplementary Figures 8-10). While not an expert in IR and RR spectroscopy, the signature of the intermediate looks rather different than the meta-F state in the wild-type (Supplementary Fig 8 C-D) and looks a lot more like the Pr state in the wild-type (Supp. Fig 9G)!?

Also the IR data show several similarities with the Pr state and hence the main conclusion from these experiments could equally be that some of the 24 substitutions apparently limit restructuring of the tongue to the classical Pr conformation, but that the remaining structural changes correspond to a somewhat impaired Pr state of Agp2!? Is this not indirectly implied by the fact that there is no difference between 273K and 240K illumination in solution?

Identifying the substitutions responsible for this impaired Pr formation could actually have far reaching implications for how the initial structural changes are communicated to the tongue and then further to the output. But for discussing the observed changes in the context of a meta intermediate, I think that the readers would benefit from more decisive data. Could UV/Vis or RR spectra of crystals (in the intermediate state) be used to support the assignment of the functional meta-F state?

5 - the authors mention the importance of the orientation of the cysteine linkage relative to biliverdin (alpha vs. beta facial). Since this cysteine is linked to a structural element that shows quite different conformations in various phytochromes it would be interesting to also show more structural details of this element - any similarities to Deinococcus, Cph1, Agp1, Idiomarina, Rhodospseudomonas, Xanthomonas, ... phytochromes? How does the beta to alpha facial transition affect the structure of the N terminal region?

6 - it is mentioned that data from crystals with different illumination times were collected. Any insights from the time dependence of structural changes?

7 - The proposed mechanism of structural propagation is interesting and important. Is the W440 conformation the same in both crystal forms (wt and variant)? Are there any other Pfr structures with tongue elements that show this feature in the absence of crystal contacts? What is the reason for observing this in Agp2 and can the authors speculate as to the potential difference with other Pfr tongue conformations. Can the pushing of Q190 onto W440 be regarded a general mechanism?

(minor) editing suggestions:

Figure 1 - please restructure the conjugated system of the biliverdins in the chemical structures presented. The authors correctly point out that the C2 methyl group is part of an sp³ hybridized carbon atom, which is incompatible with the chemical structure provided. In fact, the double bond extending towards the "vinyl"-group of BV is important for relaying the structural signal of alpha/beta-facial conjugation to the covalently attached cysteine residue.

Figure 4 - Results and Discussion mixed into the legend. Please integrate in main text.

Figure 5 - why is the Agp2-PCM structure shown for the illumination effect? Considering the different crystal lattices etc. I would encourage to show the two structural snapshots that are more easily comparable. Also showing identical orientations of the molecule in panels a and c might help the reader to more easily appreciate the structural representations.

line 598 - impact "on" the PHY tongue

line 64-65 - check grammar of this sentence (remove "could be gained")

line 169 - remove redundant "still" after adopts

Supp. Fig 10 - There is an inconsistency in the legend; If the 1657 cm⁻¹ signal reflects restructuring of the helical part of the tongue and this does not occur in the Agp2 variant, how can other signals (1642 and 1627) then indicate the refolding and degradation of the coiled structure?

Supp. Fig 11 - the rotation signs are not correct. Considering the color scheme (I would strongly recommend using other colors than black) a single rotation along a horizontal axis is sufficient.

Supp. Fig 13 - suggestion: Arg242 interacts with prop-B via a new salt bridge in meta-F

Reviewer #2 (Remarks to the Author):

This is an exciting paper that reports the crystal structure of the Meta-F intermediate of phytochrome, a widely studied photoreceptor protein that has found numerous applications in optogenetics, synthetic biology and in vivo deep-tissue imaging. Phytochromes are bistable switches that can be toggled between two functional states, Pr and Pfr, by means of light, and finding out the precise reaction cascades that connect the two states is of paramount importance. The Meta-F intermediate forms part of the Pfr to Pr reaction, and to date we only have sparse information on its molecular structure. Here, the Meta-F X-ray structure is reported, which constitutes a key step in understanding this important class of photoreceptors at the molecular level.

The paper is largely about X-ray crystallography, which is not my speciality and consequently I cannot comment on the technical aspects. The structure reveals many small-scale motions of biliverdin cofactor and nearby interacting amino acids and waters, thereby revealing important aspects of the photoactivation mechanism that take place prior to the large-scale secondary and tertiary structure motions that were shown previously to occur between Pr and Pfr.

The resonance Raman and FTIR spectroscopic work appears very solid and its conclusions with regard to the identity of the crystal-trapped intermediates are well supported.

In the discussion, the authors describe the transformations of Pfr -> Meta-F -> Pr in terms of motions of the biliverdin cofactor, amino acid side chains and internal waters. While very insightful, I miss the connection with the previously characterized Lumi-F structure(s) (Yang et al, Nature 2011), although these were determined for a different species. The paper would benefit from a more in-depth discussion on this issue.

Why is PAiRFP2 Pfr arrested at Meta-F in the crystal after illumination, while wild type appears to proceed to Pr (and leads to cracked crystals)? This is odd, given that Pr appears to be stabilized in PAiRFP as judged from Fig. S5. Is it due to the 24 substitutions with respect to wild type (which is quite a lot), or due to the different space group?

P. 4 'at higher temperatures'. Unspecific statement, what exactly do the authors mean?

Fig S10: the alpha helix-beta hairpin differential signal in phytochromes signaling refolding of the tongue was first identified for RpBphP2 by Stojkovic et al, JPCL 2014, although for the reverse reaction. I believe this should be acknowledged.

Response to Reviewer.

We thank the reviewers for their strongly positive comments and critical questions which helped us to substantially improve the manuscript. As a consequence we revised the manuscript to such an extent that only few paragraphs remained unchanged. Due to this extensive revision, marking the changes in the manuscript was not helpful. However, upon addressing the reviewers' comments below, we indicate and describe the changes made in the manuscript.

Reviewers' comments are in blue and italics and our responses are in black.

Reviewer #1 (Remarks to the Author):

The manuscript by Schmidt et al. reports crystallographic snapshots of the bathphytochrome Agp2 in its Pfr ground state as well as of (a) structural intermediate(s) along the far-red light induced conversion to the Pr state. The authors employ a combination of X-ray crystallography and spectroscopy techniques to assign this intermediate as the meta-F intermediate implicated in the conversion from Pfr to Pr. Schmidt et al. observe a set of structural rearrangements that are comparable to changes seen in canonical bacteriophytochromes featuring a Pr ground state. Eventually, the authors attempt to discuss the implications of the observed restructuring in the context of the signal transduction pathways of phytochromes in general.

This study is very well performed and the authors present their results in a conclusive way.

We thank the reviewer for this positive comment and support about the way we performed our study.

However, the short main text passage creates the impression that especially the implications of this work in the context of the phytochrome field are treated rather superficially. Also some of the conclusions from experimental data are presented in the figure legends rather than the main text, making it difficult to read the manuscript in a coherent manner. To improve the readability of the manuscript and to rule out potential inconsistencies/ambiguities in data presentation, I would recommend to work on the following points:

We agree with the reviewer. In fact, the original version was submitted as a letter to *Nature* and thus in a rather condensed format. Without these rigorous space restrictions we have now revised the manuscript completely according to the format requirements of *Nature Communications* and followed the referee's advice by including major rearrangements and expansion of the manuscript as detailed below.

1 - The authors report structures of Agp2-wt and of the PAiRFP2 variant. It is mentioned that the two proteins crystallize in different space groups and also show different oligomeric assemblies, but no visual comparison is provided. Especially since the (parallel) dimeric assembly is implicated to be important for long range signal transduction, it is puzzling that no structure of the parallel wt-Agp2 dimer is shown and consequently discussed in the context of proposed signal transduction pathways.

We have now added this comparison as a new Figure 3 showing the dimeric oligomerization of the crystallographically or symmetry-related monomers of wild-type Agp2-PCM (Fig. 3a) and Agp2-PAiRFP2 (Fig. 3b, c) in crystals and a superimposition of monomers Mol A of both structures (Fig. 3d). However, we like to note that the impact of oligomerization on the signal transduction mechanism is beyond the scope of the present study which focuses on the molecular events preceding the protein structural changes. Furthermore, crystallographic-related dimers do not provide unambiguous conclusions about functionally relevant protein-protein interactions. Nevertheless, in the revised version, we have now included a statement about the putative functional role of oligomerization in the context of a brief summary of the current view about conformational coupling between the sensor and the output module and associated signal transduction mechanism (p. 15, line 348 -).

2 - Along the line of the previous argument, the authors state in their summary that "molecular mechanisms of sensor-output communication remain elusive due to the lack of structural information of meta intermediates". This appears overstated, because the ultimate coupling with the output is achieved via the different Pr and Pfr environments.

We agree with the reviewer and modified this strong statement upon revising *Abstract* and *Introduction*.

Structural hypothesis for how these different environments are linked to the output have been put forward for the Deinococcus, Xanthomonas, Idiogramina and Rhodospseudomonas phytochrome systems where structural information for domains following the PCM is available.

These systems and sensor-output communication pathways are not discussed at all.

We have now included such information from the literature in the discussion section (p. 15, line 348 -) (see also answer point 1).

Nevertheless, structural information of meta intermediates would be highly appreciated to better understand the initial rearrangements that precede restructuring of the tongue element and how these are potentially linked to tongue reordering.

We thank the reviewer for the statement about the high impact of the present data.

3 - In order to obtain the major results of this study, a variant with 24 substitutions is used. The authors state that all of these amino acid replacements do not affect essential structural and mechanistic properties.

I would assume that the authors have tried to crystallize the variant in the same conditions as the wild-type (and vice versa).

If this has not worked, then the lack of parallel dimerization of the PAiRFP2 variant already suggests some important structural consequences.

Based on our experience with surface entropy reduction substitutions in Agp1-PCM (Nagano et al., *JBC* 2016), we assumed that Agp2-PAiRFP2 will crystallize in a different crystallization condition than wild-type Agp2-PCM. Several substitutions are located at the protein surface, which change crystallization contacts. We (crystal-) screened both constructs and found in both cases multiple crystallization conditions

resulting in crystals with very different diffraction capability. Optimization was performed for those crystallization conditions which yielded the best diffracting crystals. We like to point out that an anti-parallel crystal packing for a specific crystallization condition does not necessarily rule out a parallel dimer for other crystallization conditions, since, e.g. ions can mediate crystal contacts and therefore have an impact on the crystal packing. For example, for Agp1-PCM we could show two different crystal packings (parallel and antiparallel) with identical constructs (unpublished). Altogether, we feel that this issue could be more adequately addressed by crystallographic studies on the full-length proteins which is certainly a highly important future challenge.

As pointed out in the manuscript, the substitutions in Agp2-PAiRFP2 have some structural consequences, which are reflected, e.g. by the lower photoconversion efficacy and the enhanced fluorescence of Agp2-PAiRFP2 compared to wild-type Agp2-PCM. Crystallization-induced structural perturbations in the chromophore binding pocket can be ruled out for the Pfr state in view of the far-reaching RR spectroscopic similarities in the crystal and in solution for wild-type Agp2-PCM and the Agp2-PAiRFP2 variant. However, as revealed by RR and IR spectroscopic data, the Meta-F states of the wild-type Agp2 and Agp2-PAiRFP2 differ in some small details as the latter includes the onset of the tongue restructuring (see below #4). These structural changes are shown in Mol A of the Meta-F state, while they are partly impaired in Mol B presumably due to N-terminal contact-constraints in the crystal (Fig. 5). This is discussed in detail in the manuscript (Results section, page 9-11, Discussion section: page 11-12).

Some information with respect to the in solution oligomeric state of both proteins could be helpful to appreciate the different crystallization behavior.

The oligomerization state of full-length Agp2 (Agp2-FL) protein was found to be quite diverse. It can exist as monomer, dimer, and tetramer, and even build up multimers. The physiological and concentration-dependent oligomerization state is unknown so far. Our Agp2-PCM construct oligomerizes as dimer and tetramer in solution as found by size-exclusion chromatography (not shown). As already shown previously (Velasquez et al., Nature Chemistry 2015) the lack of the output module in Agp2-PCM affects the thermal back reaction to Pfr which is distinctly faster than in Agp2-FL. The oligomerization state of Agp2-PAiRFP2 is tetrameric in solution but in the crystal structure there is no evidence for tetramers. The structure shows tightly packed monomers that are assembled as anti-parallel dimers (new Fig. 3). From this data we conclude that it is not possible to derive an unambiguous correlation between the crystallization behavior and the oligomeric state in solution, or even the oligomerization state under physiological conditions (see above).

Also the fact that only one substitution is close to the BV environment is not correct - also M163L and A276V are positioned close to the D-ring, which is part of the central structural rearrangements. Similarly, S243N could impact the BV environment and the light induced structural rearrangements.

We thank the reviewer for this suggestion. As we mentioned in the original manuscript, V244F is directly located in the chromophore pocket, whereas M163L and S243N are part of the “third amino acid shell”. We have analyzed the interactions between chromophore and the chromophore binding site by the HBPLUS program using Ligplot (Supplementary Fig. 3 and 6). The results reveal that V244F and – with a small

contribution - A276V participate in these interactions as hydrophobic contacts. We focused our attention on V244F as this amino acid changes its conformation during photoconversion and seems to undergo an additional interaction with the propionate side chain of ring B in the Meta-F state. Presumably, this substitution accounts for the less efficient Pfr-to-Pr photoconversion and the much slower thermal back reaction to Pfr compared to wild-type Agp2-PCM. In contrast, M163L and S243N do not participate in the chromophore interaction in any of our crystal structures. These substitutions, however, might play an additional role in enhancing the fluorescence of the Agp2-PAiRFP2 construct. This view is in line with a previous analysis of NIR fluorescent phytochromes (Buhrke et al., *Sci. Reports* 2016). We have clarified this point in the revised version of the manuscript (p. 5, lines 110-113).

Original text: "This Agp2-PCM variant includes 24 substitutions, which except for Val244Phe are remote from the chromophore binding site."

Revised text: "This Agp2-PCM variant includes 24 substitutions, which except for Val244Phe and Ala267Val do not participate in the interaction between chromophore and chromophore binding site (Supplementary Fig. 3, 4 and 6)."

The most important point in this context is the observed difference with respect to the stability of illuminated crystals (wild-type crystals dissolve). The presentation of data should be more complete to enable an appreciation of the involvement of crystal lattice restraints. The contacts of the N-termini are shown and discussed but there must be more crystal contacts that stabilize the assemblies.

We understand that the expression "dissolved" is misleading. During our illumination experiments the wild-type Agp2-PCM crystals lead to non-diffracting crystals. We assume that this is a result of major photoinduced structural changes in the crystals including the conformational transition of the tongue (see e.g. Takala et al. 2014 *Nature*). The same observation during the illumination experiments can be done with the Agp2-PAiRFP2 crystals. An illumination time of more than 15 sec also leads to non-diffracting crystals. However, the Meta-F state mainly affects the chromophore and the chromophore pocket, crystal lattice restraints should not have a major impact on our observed Meta-F crystal structure. This is also in line with the IR difference spectroscopy (new Fig. 8) that shows a partial unstructuring of the tongue of Agp2-PAiRFP2 in solution consistent with the crystallographic results for the Meta-F state.

Are there differences in the tongue environments? It is stated that the wt structure does not have any contact in the tongue region but this is not explicitly stated for the variant. Could different contacts in the tongue region (in addition to the contacts of the N-termini) also be the reason for the different observations in mol A and mol B upon illumination?

Regarding to the Agp2-PAiRFP2 dimer *within the asymmetric unit* only for molecule B one van der Waals contact between P452^{MolB} and S230^{MolA} could be found (shown in new Fig. 5 (old version Supplementary Fig.11)). P452 is part of the tongue. It thus might play a role for the less unstructured tongue region of Mol B compared to Mol A. We added to the manuscript the following sentence: line 230-232):

"In Mol B, an additional hydrophobic contact in the dimer interface (Pro452B – Ser230A) is present and the segment is less unresolved than in Mol A (Fig. 5)."

Furthermore, we have also taken a closer look at the *symmetry-related, intermolecular contacts* in the crystal (using CONTACT of the CCP4 suite) which reveal the following contacts in the tongue region of molecule A for the Meta-F state of Agp2-PAiRFP2:

E468 (Mol A) - R38 (symmetry-related Mol B)
T469S (Mol A) - M39 (symmetry-related Mol B)
R471 (Mol A) - R38 (symmetry-related Mol B)
Q473 (Mol A) - M39 (symmetry-related Mol B)

The analogous analysis was made for the Pfr state of Agp2-PAiRFP2. Here the following contacts were found:

E468 (Mol A) - R38 (symmetry-related Mol B)
T469S (Mol A) - M39 (symmetry-related Mol B)
R471 (Mol A) - R38 (symmetry-related Mol B)

For both Meta-F and Pfr state no additional symmetry-related, intermolecular contacts were detected for Mol B in the tongue region.

The crystal contacts of Mol A are located at the very end of the C-terminal part of the tongue helix, and thus they are far away from the region where the light-triggered unfolding in Meta-F starts. Compared to the crystal structures of *DrBphP* in the Pfr state where the tongue is free of crystal contacts as in Agp2-PCM (Pfr), we found a very similar unfolded region.

It is currently not possible to decide, whether or not, specific contacts in the crystallographic dimer interface affect the photo-induced structural changes, particularly in the tongue region in general.

4 - The central spectroscopic data to assign the meta-F intermediate are mentioned rather superficially in the main text. Most of the results and discussion are presented in Figure legends (Supplementary Figures 8-10). While not an expert in IR and RR spectroscopy, the signature of the intermediate looks rather different than the meta-F state in the wild-type (Supplementary Fig 8 C-D) and looks a lot more like the Pr state in the wild-type (Supp. Fig 9G)!?

First, upon revising the manuscript we have now shifted a major part of the RR and IR spectroscopic analysis from the *Supplementary Information* to the main manuscript, including two figures (Fig. 4, Fig. 8) (see revised manuscript, line 139 - and line 233 -). We also agree with the reviewers that the original presentation of the RR data (original Supplementary Fig. 7) was not very clear. In the revised Supplementary Fig. 9 and its caption (and also in Supplementary Fig. 7) we show more clearly that the RR spectrum of Meta-F state of Agp2-PAiRFP2 exhibits a vibrational band pattern that is very similar to that of the Meta-F state of wild-type Agp2. However, there are notable differences specifically for two modes (C-D stretching and N-H in-plane bending) that are more closely related to the spectrum of the Pr state of wild-type full-length Agp2, indicating subtle differences in the C-D methine bridge geometry and different hydrogen bonding interactions of rings B and C.

Also the IR data show several similarities with the Pr state and hence the main conclusion from these experiments could equally be that some of the 24 substitutions apparently limit restructuring of the tongue to the classical Pr conformation, but that the remaining structural changes correspond to a somewhat impaired Pr state of

Agp2!? Is this not indirectly implied by the fact that there is no difference between 273K and 240K illumination in solution?

This is correct and we have to admit that this was not clearly described in the original version of the manuscript. Altogether, the RR and IR spectra indicate the high structural similarity of the chromophore in the Meta-F state of Agp2-PAiRFP2 and wild-type Agp2. However, unlike to wild-type Agp2-PCM in solution, the Meta-F state of Agp2-PAiRFP2 displays the onset of the structural transformation of the tongue, documented by both the Meta-F crystal structure and the IR difference spectrum (after irradiation of 273 K in solution). The IR difference spectrum of irradiated Agp2-PAiRFP2 constitutes the final product of the phototransformation of Pfr. The only deviation between IR spectroscopic and crystallographic data of Agp2-PAiRFP2 refers to the formation of the small β -hairpin segment detected in the solution IR spectra which may be impaired or not fully reached in the crystal. Nevertheless, the good overall agreement of the crystallographic and IR spectroscopic data may also indicates that Agp2-PAiRFP2 in solution is not capable to perform the complete structural transformation of the tongue. (line 233 - , lines 250 - 262).

Identifying the substitutions responsible for this impaired Pr formation could actually have far reaching implications for how the initial structural changes are communicated to the tongue and then further to the output. But for discussing the observed changes in the context of a meta intermediate, I think that the readers would benefit from more decisive data. Could UV/Vis or RR spectra of crystals (in the intermediate state) be used to support the assignment of the functional meta-F state?

We agree to the reviewer, that for spectroscopically classifying the intermediate states of illuminated crystals, RR spectra on the crystals are the most decisive data. We added a new Fig. 4 and Supplementary Fig. 7. (and captions) with a comparison of RR spectra from Agp2-PAiRFP2 (crystals and solution) in Meta-F and Pfr states.

A source of confusion and ambiguity in assigning intermediates of the phototransformation of phytochromes roots in the underlying technique. UV-vis absorption probes the chromophore and thus can hardly distinguish between states of related chromophore structures. Thus, although UV-vis absorption spectra of Agp2-PAiRFP2 show the typical Pfr/Pr conversion pattern, the protein changes of the final photoproduct deviate from that of the wild-type full-length protein (as seen by IR spectroscopy). If we define the Pr state as that state in which chromophore and protein structural changes of the Pfr phototransformation have been completed, IR spectroscopy allows for an unambiguous conclusion which can also account for the RR spectroscopic results (as a significantly more sensitive technique to identify the chromophore state compared to UV-vis spectroscopy). Accordingly, we can safely assigned the crystallized intermediate to a precursor of Pr, i.e. more precisely to a Meta-F substate with a chromophore structure largely similar (albeit not identical) to the wild-type Meta-F state but including the onset of the restructuring of the tongue. We hope that with the extended revision of the manuscript (including the new and revised figures in the manuscript and the Supplementary Information) we have clarified the characterization and assignment of the Meta-F state.

5 - the authors mention the importance of the orientation of the cysteine linkage relative to biliverdin (alpha vs. beta facial). Since this cysteine is linked to a structural element that shows quite different conformations in various phytochromes it would be interesting to also show more structural details of this element - any similarities to

Deinococcus, Cph1, Agp1, Idiominarina, Rhodopseudomonas, Xanthomonas, ... phytochromes? How does the beta to alpha facial transition affect the structure of the N terminal region?

In all known (bacterial) structures the thioether linkage between the chromophore and cysteine is (nearly) α -facial in Pr and β -facial in Pfr. (Note that all available crystal structures have different resolutions and content of radiation damage in this region.) For Cph1 and Arabidopsis PhyB the chromophore binding cysteine is located in the GAF domain. Therefore, these structures have been excluded from this comparison. The structure of the N-terminal region seems to be very diverse in the different organisms, but the orientation of the cysteine with respect to the chromophore is consistent. For Agp2-PAiRFP2 (Mol A) we observed a movement of and conformational change in the N-terminus appearing from the very beginning up to Arg15^{MolA}. After illumination of the crystals the new conformation of the N-terminus in Mol A is very similar to the N-terminus of Agp1-PCM in Pr state.

The Pfr structure of *Deinococcus radiodurans* DrBphP-PCM shows a shifted N-terminus that has a high similarity to wild-type Agp2-PCM and Agp2-PAiRFP2 in their Pfr states. The new position of the cysteine is located β -facially to the chromophore. In the Pr state of DrBphP we can observe a higher flexibility of the cysteine attachment to the chromophore which can be subdivided in two groups. But in both groups cysteine binding is oriented α -facial to the chromophore.

A more detailed description of the β/α -facial transition is now included in the revised version of the manuscript (line 217-222; 295-300; 332-336).

Additionally to Fig. 7, we added a new Supplementary Fig. 11 to the manuscript that displays the orientation of cysteine binding to the bilin-type chromophore of different phytochromes.

6 - it is mentioned that data from crystals with different illumination times were collected. Any insights from the time dependence of structural changes?

When using illumination times below 3 sec data sets of the Pfr state were obtained. Illumination times longer than 15 sec lead to non-diffracting crystals such that we considered 3 sec illumination time as the optimum condition to produce Meta-F crystals with a valuable diffraction quality.

7 - The proposed mechanism of structural propagation is interesting and important. Is the W440 conformation the same in both crystal forms (wt and variant)?

W440 adopts a similar conformation in the Pfr states of wild-type Agp2-PCM and Agp2-PAiRFP2.

Are there any other Pfr structures with tongue elements that show this feature in the absence of crystal contacts? What is the reason for observing this in Agp2 and can the authors speculate as to the potential difference with other Pfr tongue conformations. Can the pushing of Q190 onto W440 be regarded a general mechanism?

As mentioned in the manuscript PaBphP2 structure (PDB entry 3NHQ; 2.55 Å resolution) is the crystal structure with the highest resolution of a previously published Pfr bathy phytochrome structures (Of note, i.e. the bacteriophytochrome from

Xanthomonas campestris (PDB entry 5AKP) is also a bathy-like phytochrome, but the crystal structure has a low resolution (below 3.2 Å) and a Pfr/Pr-mixed state). In the *PaBphP* structure W437 (W440 in Agp2-PCM or Agp2-PAiRFP2) is part of a β -strand from the tongue and tightly packed against a β -strand of another crystallographic monomer within the unit cell resulting in a β -sheet formation. Consequently, the location and conformation of W437 in the *PaBphP* crystal structure is more or less biased by crystal packing and might be different from that in solution. Remarkably, Q188 of *PaBphP* shows high conformational similarity to Q190 in Agp2-PCM or Agp2-PAiRFP2 (but without a water molecule between Gln and ring D of BV).

Another Pfr structure showing this conserved tryptophan (W440 in Agp2 nomenclature) is the crystal structure of the prototypical phytochrome *DrBphP* (PDB entry: 5C5K; 3.31 Å). To gain the light-activated Pfr-like state a single substitution F469W (part of the tongue) was introduced by Burgie and coworkers. The protein was illuminated in solution and subsequently crystallized. In this structure W451, equivalent to W440 in Agp2-PCM or Agp2-PAiRFP2, was resolved in three out of the four molecules in the asymmetric unit. It shows in three monomers different conformations and also a different distance (3.5 - 4.9 Å) to H201 (which is equivalent position as Q190 in Agp2-PCM). For all four *DrBphP* molecules in this light-activated Pfr-like state the tongue regions were not completely resolved. Besides for molecule A, no crystal contacts stabilize these tongue regions (p.14, line: 338 -).

Highly flexible regions (unresolved regions in *DrBphP*: Mol A 454-459, Mol B 452-459, Mol C 450-462, Mol D 452-463) comparable with those in our Meta-F structure of Agp2-PAiRFP2 (Mol A 439-448, Mol B 439-442) strengthen our proposed mechanism that the light-triggered restructuring of the tongue region starts (Pfr/Pr photoconversion) within this region and propagates to the rest of the tongue.

Furthermore, the partially unstructured tongue in this crystal structure of *DrBphP* (also shown for the light-illuminated crystal structure of *DrBphP*, Takala et al., *Nature* 2014; PDB entry 4O01) indicates that Pfr has not been reached completely. Another structural feature that supports our proposed mechanism is the conformational switch of H201 in *DrBphP*. Burgie and coworkers already discussed in their paper (Burgie et al., *Structure* 2016) the light-dependent conformational change of H201 which corresponds to the conformational change of Q190, with is in the analogue position in Agp2-PAiRFP2. Both amino acids display an outward shift towards the tongue region (Pfr/Pr photoconversion).

These relationships with other phytochromes are now discussed in the Discussion section of the revised manuscript.

(minor) editing suggestions:

Figure 1 - please restructure the conjugated system of the biliverdins in the chemical structures presented. The authors correctly point out that the C2 methyl group is part of an sp³ hybridized carbon atom, which is incompatible with the chemical structure provided. In fact, the double bond extending towards the "vinyl"-group of BV is important for relaying the structural signal of alpha/beta-facial conjugation to the covalently attached cysteine residue.

We changed the figure accordingly as suggested.

Figure 4 - Results and Discussion mixed into the legend. Please integrate in main text.

We followed the reviewer's suggestion in the revised version of the manuscript.

Figure 5 - why is the Agp2-PCM structure shown for the illumination effect? Considering the different crystal lattices etc. I would encourage to show the two structural snapshots that are more easily comparable. Also showing identical orientations of the molecule in panels a and c might help the reader to more easily appreciate the structural representations.

We followed the reviewer's suggestion in the revised version of the manuscript and show in figure 9 (5 in the original version) Agp2-PAiRFP2 for representing the Pfr state.

line 598 - impact "on" the PHY tongue

Done.

line 64-65 - check grammar of this sentence (remove "could be gained")

Done.

line 169 - remove redundant "still" after adopts

Done.

Supp. Fig 10 - There is an inconsistency in the legend; If the 1657 cm⁻¹ signal reflects restructuring of the helical part of the tongue and this does not occur in the Agp2 variant, how can other signals (1642 and 1627) then indicate the refolding and degradation of the coiled structure?

We have shifted this figure in the main text as Fig. 8. The 1657-cm⁻¹ band originates from the helical part of the tongue which is only refolded in the transition to Pr of the wild-type full-length Agp2. The 1627 cm⁻¹ band is associated with the coil part of the tongue which in the crystal structure of Meta-F is partially not resolved, consistent with its degradation as revealed by the IR difference spectrum of Agp2-PAiRFP2. We have no indication for the formation of β -sheet structure in Agp2-PAiRFP2 that would lead to a positive band at ca. 1625 – 1630 cm⁻¹, thereby compensating the negative "coil" band in wild-type full-length Agp2. The positive 1642 cm⁻¹ band just refers to the small β -hairpin segment that is formed in solution but may be impaired or not fully reached in the crystal. In the revised version, these assignments and the corresponding interpretations are described in more detail (line 233 – 262).

Supp. Fig 11 - the rotation signs are not correct. Considering the color scheme (I would strongly recommend using other colors than black) a single rotation along a horizontal axis is sufficient.

We changed the figure accordingly. We shifted the figure in the main manuscript as new Fig.5.

*Supp. Fig 13 - suggestion:
Arg242 interacts with prop-B via a new salt bridge in meta-F*

Done.

Reviewer #2
(Remarks to the Author):

This is an exciting paper that reports the crystal structure of the Meta-F intermediate of phytochrome, a widely studied photoreceptor protein that has found numerous applications in optogenetics, synthetic biology and in vivo deep-tissue imaging. Phytochromes are bistable switches that can be toggled between two functional states, Pr and Pfr, by means of light, and finding out the precise reaction cascades that connect the two states is of paramount importance. The Meta-F intermediate forms part of the Pfr to Pr reaction, and to date we only have sparse information on its molecular structure. Here, the Meta-F X-ray structure is reported, which constitutes a key step in understanding this important class of photoreceptors at the molecular level.

The paper is largely about X-ray crystallography, which is not my speciality and consequently I cannot comment on the technical aspects. The structure reveals many small-scale motions of biliverdin cofactor and nearby interacting amino acids and waters, thereby revealing important aspects of the photoactivation mechanism that take place prior to the large-scale secondary and tertiary structure motions that were shown previously to occur between Pr and Pfr.

The resonance Raman and FTIR spectroscopic work appears very solid and its conclusions with regard to the identity of the crystal-trapped intermediates are well supported.

We thank the reviewer for this positive comment and support about the way we performed our study.

In the discussion, the authors describe the transformations of Pfr -> Meta-F -> Pr in terms of motions of the biliverdin cofactor, amino acid side chains and internal waters. While very insightful, I miss the connection with the previously characterized Lumi-F structure(s) (Yang et al, Nature 2011), although these were determined for a different species. The paper would benefit from a more in-depth discussion on this issue.

We have now included these results in the discussion section on p. 12, line 281-.

Why is PAiRFP2 Pfr arrested at Meta-F in the crystal after illumination, while wild type appears to proceed to Pr (and leads to cracked crystals)? This is odd, given that Pr appears to be stabilized in PAiRFP as judged from Fig. S5. Is it due to the 24 substitutions with respect to wild type (which is quite a lot), or due to the different space group?

As detailed explained in the answer for reviewer 1, Agp2-PAiRFP2 crystals also lead to non-diffracting crystals but after an illumination time of more than 15 sec, whereas in Agp2-PCM crystals this appears much faster. Therefore, this behavior indicate that the Agp2-PAiRFP2 crystals are not arrested in this Meta-F substate.

P. 4 'at higher temperatures'. Unspecific statement, what exactly do the authors mean?

Original text:

“The Pfr Agp2-PCM crystals cracked immediately after irradiation with far-red light at higher temperatures indicating major protein conformational changes in the crystal.”

Illumination experiments on the crystals at near room temperature performed at the beamline with the method of fast interruption of the nitrogen cooling stream (at 100K) and parallel illumination for a short time lead to non-diffracting crystals. By this method an exact temperature cannot be determined. We would assume that interruption of the nitrogen cooling would lead to ambient temperature on and within the crystals.

Therefore, we specified the sentence to:

“The Pfr Agp2-PCM crystals lose their diffraction capability immediately after irradiation with far-red light at ambient temperatures indicating major protein conformational changes in the crystal.” (line 96-)

Fig S10: the alpha helix-beta hairpin differential signal in phytochromes signaling refolding of the tongue was first identified for RpBphP2 by Stojkovic et al, JPCL 2014, although for the reverse reaction. I believe this should be acknowledged.

We included the new reference of RpBphP2 (Stojkovic et al, JPCL 2014).

Reviewers' comments:

Reviewer #1 (Remarks to the Author):

My overall impression is that the manuscript has greatly improved by the changes made by the authors. Many aspects related to my previous concerns have been adequately addressed and I really enjoyed reading the article.

However, there remains one impression that I cannot really overcome and that somewhat complicates the final assessment. This relates to the assignment of THE intermediate as THE meta-F state.

As the authors state themselves, there are some differences in RR spectra between meta-F of the variant and the wild-type construct. Also the two protomers of the crystallographic dimer are not identical and provide different structural information - potentially for intermediates along the path to Pr. Therefore, I was wondering if one should reconsider the assignment to meta-F and relativize this to state that different intermediates along the path to Pr can be observed. Whether any of the two is THE meta-F intermediate is in my opinion not so easy to state, also based on the spectra that the authors provide. A toned down argument that the mixture of the two (or one of the) intermediates in the crystals have meta-F like properties could then be used in the discussion and all subsequent conclusions and hypotheses could remain unchanged and are in my opinion very interesting.

The detail/aspect that makes me feel queasy, is the fact that the meta-F RR spectra of the variant can be acquired at 273 K where the corresponding UV/Vis spectra look a lot like a standard Pr state. As the authors correctly state in their replies, the information content of UV/Vis spectra is limited and therefore potentially difficult to discuss, but showing a Pr-like UV/Vis spectrum and stating that under the same conditions you acquire a meta-F like RR spectrum is somewhat difficult to understand.

Taken together, I would tone down the meta-F assignment and state that two interesting crystal-trapped intermediates along the path to Pr can be observed in this 24-substitution variant. As the authors state themselves, this variant appears not to refold the alpha-helical tongue part to a b-sheet; maybe one of the mutations enables a Pr-like stabilization without tongue refolding? In this respect UV/Vis spectra of the crystals and a comparison to the in solution data would have been informative; can a Pr-like spectrum be observed without a Pr-like tongue?

minor points:

line 46 - PAS domain abbreviation - ARNT not Arndt

line 171 - conditions, showed some spectral variation

line 234 - please rephrase "less unresolved"

line 261 - may also indicate

Figure 1 - the conjugated double bond system was corrected in the upper chemical structure but not in the ZZZssa conformation

Ala267Val vs Ala276Val - 276 should be correct but sometimes the authors refer to 267 - eg line 111 and other instances

Supp-Figure 5 - half-time is something I know from soccer; either half-live or lifetime should be used correctly here.

Supp-Figure 11 - panels f and I have inverted orientations/views to the remaining panels

Supp table 1 - should the tilt angles between rings C and D not account for the $\sim 180^\circ$ rotation - more difference would be expected for Pr vs Pfr? some definition as to where e.g. the carbonyl points might serve as a orientation point.

Response to the Reviewer.

We thank the reviewer for his/her positive comments. The remaining critical points are addressed below (Reviewer's comments are displayed in blue and italics and our response in black). Changes made in the 2nd revised manuscript are highlighted in yellow.

Reviewer #1 (Remarks to the Author):

My overall impression is that the manuscript has greatly improved by the changes made by the authors. Many aspects related to my previous concerns have been adequately addressed and I really enjoyed reading the article.

However, there remains one impression that I cannot really overcome and that somewhat complicates the final assessment. This relates to the assignment of THE intermediate as THE meta-F state.

1. As the authors state themselves, there are some differences in RR spectra between meta-F of the variant and the wild-type construct. Also the two protomers of the crystallographic dimer are not identical and provide different structural information - potentially for intermediates along the path to Pr. Therefore, I was wondering if one should reconsider the assignment to meta-F and relativize this to state that different intermediates along the path to Pr can be observed. Whether any of the two is THE meta-F intermediate is in my opinion not so easy to state, also based on the spectra that the authors provide. A toned down argument that the mixture of the two (or one of the) intermediates in the crystals have meta-F like properties could then be used in the discussion and all subsequent conclusions and hypotheses could remain unchanged and are in my opinion very interesting.

We fully agree with the reviewer and regret that we did not clarify this point already in the previous revision. Just the fact that there are two "sub-states" in the crystal structure implies that one cannot talk about "THE meta-F state". Instead, it would be quite surprising if the reaction pathway from Pfr to Pr involved only two minima (Lumi-F, Meta-F). We have therefore corrected the notation throughout the text and pointed out that "Meta-F" includes in fact sub-states. These changes were made at lines 33, 68, 153, 157, 171, 195/196, 227, 251, 261, 266 and 346 and in the figure legends (Fig. 2,5 and 6) and the Supplementary Information figure legends (Si-Fig. 1,2,6 and 11)

2. The detail/aspect that makes me feel queasy, is the fact that the meta-F RR spectra of the variant can be acquired at 273 K where the corresponding UV/Vis spectra look a lot like a standard Pr state. As the authors correctly state in their replies, the information content of UV/Vis spectra is limited and therefore potentially difficult to discuss, but showing a Pr-like UV/Vis spectrum and stating that under the same conditions you acquire a meta-F like RR spectrum is somewhat difficult to understand. Taken together, I would tone down the meta-F assignment and state that two interesting crystal-trapped intermediates along the path to Pr can be observed in this 24-substitution variant. As the authors state themselves, this variant appears not to refold the alpha-helical tongue part to a b-sheet; maybe one of the mutations enables a Pr-like stabilization without tongue refolding? In this respect UV/Vis spectra of the crystals and a comparison to the in solution data would have been informative; can a Pr-like spectrum be observed without a Pr-like tongue?

The RR spectra of Pr (keto-form) and Meta-F differ only in subtle details implying that the structure of the chromophore and the chromophore binding pocket are essentially the same in both states (Velasquez et al., Nat. Chem. 7, 423-430, 2015). The similar conclusion can be drawn for the Pr and Meta-F states of plant phytochrome (Matysik et al., Biochemistry 34, 10497-10507, 1995). Since the structure of the chromophore and its interactions with the immediate protein environment are the key determinants for the UV-vis absorption spectrum it is not surprising that also the UV-vis absorption spectra of Pr and Meta-F are very similar (Eilfeld & Rüdiger, Z. Naturforsch. 40c, 109 — 114, 1985). On the other hand, the main structural change that takes place during the transition from Meta-F to Pr is evidently the restructuring of the tongue, which is only reflected by the IR spectra. Accordingly, one may describe the relationship between Meta-F and Pr by the simplified formula: Pr = Meta-F + restructured tongue. Correspondingly, the answer to the reviewer's question "*can a Pr-like spectrum be observed without a Pr-like tongue?*" is yes, but such a spectrum corresponds to the Meta-F state. Thus, UV-vis spectroscopy would certainly not be the adequate method to find out if there is a critical single mutation in Agp2-PAiRFP that impairs the tongue restructuring.

The above considerations also illustrate the limitations of the widely accepted distinction of the photocycle intermediates in terms of their UV-vis spectra which are not suitable to capture and describe all minima of the potential energy surface for the photoinduced transformations in phytochromes. In this respect, we fully agree with the reviewer's request to avoid talking about THE Meta-F state (which we have now revised, see #1). We also agree with the reviewer that the caption of the Supporting Figure S5 is confusing. We have therefore revised the caption by pointing out the similarities with the spectra of a protonated ZZZssa chromophore in phytochromes, i.e. Meta-F and Pr.

minor points:

line 46 - PAS domain abbreviation - ARNT not Arndt

corrected

line 171 - conditions, showed some spectral variation

corrected

line 234 - please rephrase "less unresolved"

corrected

line 261 - may also indicate

corrected

Figure 1 - the conjugated double bond system was corrected in the upper chemical structure but not in the ZZZssa conformation

As our crystal structure (6G20) shows a *ZZZssa* conformation for the chromophore but does not display an exocyclic orientation of C2 at ring A, we did not change the conjugated double bond system for the *ZZZssa* conformation in the chemical structure of Figure 1.

Ala267Val vs Ala276Val - 276 should be correct but sometimes the authors refer to 267 - eg line 111 and other instances

corrected

Supp-Figure 5 - half-time is something I know from soccer; either half-live or lifetime should be used correctly here.

corrected

Supp-Figure 11 - panels f and l have inverted orientations/views to the remaining panels

All panels in Suppl. Fig 11 are superimposed, thus have the same orientation. All crystal structure superpositions of backbone α -carbon traces were performed using the CCP4 program LSQKAB.

Supp table 1 - should the tilt angles between rings C and D not account for the $\sim 180^\circ$ rotation - more difference would be expected for Pr vs Pfr? some definition as to where e.g. the carbonyl points might serve as a orientation point.

As the reviewer states from the tilt angles given in Suppl. Table 1 it is impossible to tell if one ring is rotated either clockwise or anti-clockwise with respect to the other (φ vs $-\varphi$) or if a ring is "flipped" or not within a plane (φ vs $(180^\circ - \varphi)$). All angles measured by the method implemented in UCSF Chimera are between 0° and 90° , which means that (1) they are only positive and (2) cannot distinguish between flipped and non-flipped orientations. In order to ensure that the information provided in the table is unambiguous and as complete as possible, we defined the configuration/conformation of the chromophore in each structure (*ZZZssa* or *ZZEssa*) and stated that ring D is in α -facial disposition with respect to the remaining portion of the chromophore in all the structures included in the analysis. The corresponding situation is nicely illustrated in Fig 4. of N.C. Rockwell et al., PNAS 106, 6123-6127 (2009).

We believe that this description, which has also been used by other groups in previous publications on phytochrome structures, is currently the most practicable. Clear distinction between the different scenarios mentioned above would require additional efforts in precisely defining (1) the sign of a tilt angle (such that the sign would change if the order of the planes is swapped) and (2) the direction (not just the orientation) of the normal vectors of each plane.